# Equivariant Spherical Transformer for Efficient Molecular Modeling

## Abstract

Equivariant Graph Neural Networks (GNNs) have significantly advanced the modeling of 3D molecular structure by leveraging group representations. However, their message passing, heavily relying on Clebsch-Gordan tensor product convolutions, suffers from restricted expressiveness due to the limited non-linearity and low degree of group representations. To overcome this, we introduce the Equivariant Spherical Transformer (EST), a novel plug-and-play framework that applies a Transformer-like architecture to the Fourier spatial domain of group representations. EST achieves higher expressiveness than conventional models while preserving the crucial equivariant inductive bias through a uniform sampling strategy of spherical Fourier transforms. As demonstrated by our experiments on challenging benchmarks like OC20 and QM9, EST-based models achieve state-of-the-art performance. For the complex molecular systems within OC20, small models empowered by EST can outperform some larger models and those using additional data. In addition to demonstrating such strong expressiveness, we provide both theoretical and experimental validation of EST's equivariance as well, paving the way for new research in this area. Our code is anonymously released in https://anonymous.4open.science/r/EST-1976

## 1 Introduction

Graph neural networks (GNNs) are increasingly used for modeling molecular systems and approximating quantum mechanical calculations, providing crucial support for computational chemistry tasks like drug discovery (Senior et al., 2020; Kovács et al., 2025; Deng et al., 2023) and material design (Zitnick et al., 2020). Compared to traditional methods such as Density Functional Theory (DFT), GNNs can predict quantum properties in fractions of a second, significantly reducing computational cost from hours or days. Among the various GNN architectures, SE(3)-equivariant GNN (Thomas et al., 2018; Brandstetter et al., 2022) has become a leading variant due to their inherent physical constraints, which can effectively capture intricate atomic interactions and address key challenges in molecular modeling.

In molecular systems, SE(3)-invariance and SE(3)-equivariance are fundamental constraints. For instance, molecular energies remain constant under rotation, while atomic forces rotate in concert with the molecule equivariantly. Early invariant models, such as SchNet (Schütt et al., 2018) and HIP-NN (Lubbers et al., 2018), rely on interatomic distances in their message-passing blocks, consequently limiting their ability to capture interactions involving triplets or quadruplets of atoms (Miller et al., 2020). Directional GNNs (Gasteiger et al., 2020; 2021) address this problem by explicitly incorporating bond angles and dihedral angles. While these models showed performance improvements, their expressiveness remains constrained due to a heavy reliance on handcrafted features. To capture deeper features, SE(3)-equivariant GNNs employ group representations as node embeddings and construct steerable message-passing blocks (Thomas et al., 2018; Brandstetter et al., 2022; Liao & Smidt, 2023), where tensor product are applied to capture equivariant interactions between group embeddings (see Figure 1(b)). These GNNs achieve improved performance through end-to-end modeling, eliminating the need for hand-crafted features. Nevertheless, tensor product operations inherently suffer from limited non-linearity (Thomas et al., 2018), and their expressiveness is bounded by the degree of group representations (Dym & Maron, 2021; Joshi et al., 2023; Cen et al., 2024). An alternative equivariant approach is to capture features in the spatial domain by applying a Fourier Transform (FT) to group representations, where embeddings are defined by square-integrable

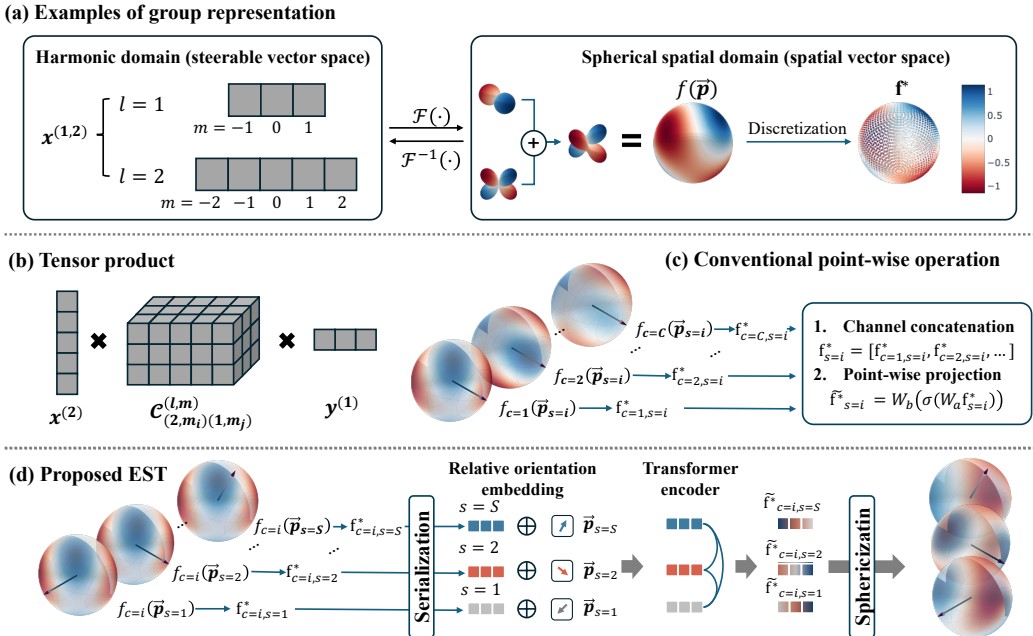

Figure 1: **Overview of operations in the spherical spatial domain.** (a) Examples of group representation. A Fourier transform is applied to project steerable vectors onto the spatial domain, which are then stored through sampling (i.e., discretization). (b) Tensor product operation. Steerable vectors are combined using the Clebsch-Gordan coefficient $\mathcal{C}$. (c) Conventional point-wise operation. All channels within the same orientation are processed by a vanilla neural network ((with linear projection $W$ and activation $\sigma(\cdot)$), and no interaction is performed across different orientations. (d) Proposed EST operation. A Transformer-based structure captures dependencies across all orientation pairs to model complex atomic behavior.

spherical functions (see the right side of Figure 1(a)). Prior works (Zitnick et al., 2022; Passaro & Zitnick, 2023; Liao et al., 2024b; Fu et al., 2025) typically apply simple point-wise neural networks to extract features from these spherical functions, operating independently on each orientation (see Figure 1(c)). However, **operations introducing non-linearity across different orientations in the spatial domain are rarely considered, as they risk violating equivariance. We demonstrate that this challenge can be overcome by subtly redesigning the FT sampling and architecture, which can also provide additional expressiveness** (Further analysis is provided in Section 3.2).

This paper presents the Equivariant Spherical Transformer (EST), a plug-and-play equivariant framework for high-fidelity modeling of atomic interactions. Our methodology initiates the process in the Fourier spatial domain of group representations by converting the spatial representation into a sequence of points. Subsequently, a Transformer-based architecture, as depicted in Figure 1(d), is employed to capture dependencies across different orientations on this sequence. A key finding of our work is that conventional spherical FT sampling methods, exemplified by the e3nn grid (Geiger et al., 2022), significantly degrade equivariance. To mitigate this issue, we introduce a new, provably spherical sampling strategy grounded in the Fibonacci lattice, which we validate through theoretical and empirical analyses. Additionally, we incorporate a "hybrid mixture of experts" structure within our Transformer, featuring feed-forward networks in both the spatial and harmonic domains to strike an optimal balance between equivariance and model capacity. Experiments demonstrate that EST-based architectures attain state-of-the-art (SOTA) performance on standard benchmarks such as OC20 (Chanussot et al., 2021) and QM9 (Ramakrishnan et al., 2014), with stable and notable improvements. On the complex molecular systems of the OC20 dataset, a compact EST model surpasses the performance of larger models and those leveraging external training data, thereby attesting to its strong expressiveness for modeling molecular systems.

**Our contributions are:** (i) We propose EST, a plug-and-play framework whose expressiveness subsumes that of traditional tensor products-based operations. (ii) We guarantee the equivariance of EST by subtly employing an approximately uniform sampling implementation on spherical FT. (iii) We demonstrate the superior performance of EST through extensive experiments. Furthermore, our

ablation studies confirm the equivariance and enhanced expressiveness of EST, establishing it as a promising building block for a wide range of equivariant architectures.

## 2 PRELIMINARIES AND RELATED WORK

In this section, we discuss related work and the corresponding mathematical background relevant to equivariant GNNs. We begin by listing the notations frequently used throughout the paper. We denote the unit sphere as $\mathbb{S}^2$, where the coordinate of a spherical point (or orientation) $\vec{\mathbf{p}} = (\theta, \varphi)$ is represented by its polar angle $\theta$ and azimuth angle $\varphi$. The symbol $\mathbb{R}$ represents the set of real numbers, while $\mathbf{R}$ denotes a rotation matrix for 3D vectors. We use $[\cdot, \cdot]$ or $\oplus$ to indicate tensor concatenation and $\tilde{\ }$ to represent the update of tensors, respectively.

**Message Passing Neural Networks**  Consider a graph $\mathcal{G} = (\mathcal{V}, \mathcal{E})$, with nodes $v_i \in \mathcal{V}$ and edges $e_{ij} \in \mathcal{E}$. Each node $v_i$ has an embedding $\mathbf{x}_i$ and an attribute $\mathbf{z}_i$, and each edge $e_{ij}$ has an attribute $\mathbf{a}_{ij}$. The Message Passing Neural Network (MPNN) (Schütt et al., 2018), a specific type of GNN, updates node embeddings through a message block $\mathbf{M}(\cdot)$ and an update block $\mathbf{U}(\cdot)$ via the following steps:

$$\mathbf{m}_{ij} = \mathbf{M}(\mathbf{x}_i, \mathbf{x}_j, \mathbf{z}_i, \mathbf{z}_j, \mathbf{a}_{ij}, \vec{\mathbf{r}}_{ij}), \quad \text{and} \quad \tilde{\mathbf{x}}_i = \mathbf{U}(\mathbf{x}_i, \sum_{j \in \mathcal{N}(i)} \mathbf{m}_{ij}), \tag{1}$$

where we use $\vec{\mathbf{r}}_{ij}$ to denote the relative position of node $i$ and node $j$ in 3D space, and the neighborhood $\mathcal{N}(i)$ is typically defined by a cutoff radius: $\mathcal{N}(i) = \{j \mid \|\vec{\mathbf{r}}_{ij}\| \leq r_{cut}\}$. The node attribute $\mathbf{z}_i$ contains atomic information such as the atomic type, and the edge attribute $\mathbf{a}_{ij}$ contains atomic pair information such as distance and bond type. By stacking multiple message and update blocks, the final node embeddings can be used to model atomic interactions or represent molecular properties.

**Equivariance and Invariance**  Given a group $\mathbf{G}$ and a transformation parameter $g \in \mathbf{G}$, a function $\phi : \mathcal{X} \to \mathcal{Y}$ is said to be equivariant to $g$ if it satisfies:

$$\phi(T(g)[x]) = T'(g)[\phi(x)], \tag{2}$$

where $T'(g) : \mathcal{Y} \to \mathcal{Y}$ and $T(g) : \mathcal{X} \to \mathcal{X}$ denote the corresponding transformations on $\mathcal{Y}$ and $\mathcal{X}$, respectively. Invariance is a special case of equivariance where $T'(g)$ is the identity transformation. In this paper, we primarily focus on SO(3) equivariance, i.e., equivariance under 3D rotations, as it is closely related to the interactions between atoms in molecules [1].

**Spherical Harmonics and Steerable Vectors**  Spherical harmonics, a set of orthonormal basis functions defined over the sphere $\mathbb{S}^2$, are commonly employed in equivariant models. The real-valued spherical harmonics are typically denoted as $\{Y^{(l,m)} : \mathbb{S}^2 \to \mathbb{R}\}$, where $l$ represents the degree and $m$ represents the order. For any orientation $\vec{\mathbf{p}}$, we define $\mathbf{Y}^{(l)}(\vec{\mathbf{p}}) = [Y^{(l,-l)}(\vec{\mathbf{p}}), Y^{(l,-l+1)}(\vec{\mathbf{p}}), ..., Y^{(l,l)}(\vec{\mathbf{p}})]$, a vector of size $2l + 1$, and $\mathbf{Y}^{(0 \to l)}(\vec{\mathbf{p}}) = [\mathbf{Y}^{(0)}(\vec{\mathbf{p}}), ..., \mathbf{Y}^{(l)}(\vec{\mathbf{p}})]$, a vector of size $(l + 1)^2$.

A key property of spherical harmonics is their behavior under rotation $\mathbf{R} \in SO(3)$:

$$\mathbf{Y}^{(l)}(\mathbf{R}\vec{\mathbf{r}}) = \mathbf{D}^{(l)}(\mathbf{R})\mathbf{Y}^{(l)}(\vec{\mathbf{r}}), \tag{3}$$

where $\mathbf{D}^{(l)}(\mathbf{R})$ is a $(2l + 1) \times (2l + 1)$ matrix known as the Wigner-D matrix of degree $l$. Notably, $\mathbf{D}^{(1)}(\mathbf{R}) = \mathbf{R}$. Thus, $\mathbf{R}$ and $\mathbf{D}^{(l)}(\mathbf{R})$ can represent $T(g)$ and $T'(g)$ in equation 2. Following the convention in (Chami et al., 2019; Brandstetter et al., 2022), we say that $\mathbf{Y}^{(l)}(\vec{\mathbf{p}})$ is steerable by the Wigner-D matrix of the same degree $l$. Furthermore, a vector that transforms according to an $l$-degree Wigner-D matrix is termed an $l$-degree steerable vector or a type-$l$ vector, residing in the vector space $\mathbb{V}_l$. Further mathematical details are available in Appendix B.1.

**Equivariant architectures**  A common practice in equivariant models is to use steerable vectors as node embedding and encode relative positions $\vec{\mathbf{r}}_{ij}$ using spherical harmonics as edge attributes. TFN (Thomas et al., 2018) and NequIP (Batzner et al., 2022) provide a general framework that learns the interaction between node embeddings and edge attributes through equivariant convolution filters. These filters are composed of equivariant operations such as degree-wise linear layers, Clebsch-Gordan (CG) tensor products, and Gate mechanisms. SEGNN (Brandstetter et al., 2022) extends this convolution framework by more non-linear operations to achieve enhanced learning ability. SE(3)-Transformer (Fuchs et al., 2020) and Equiformer (Liao & Smidt, 2023) introduce an attention mechanism that assigns rotation-invariant weights to edge attributes, thereby improving the learning of

---

[1]Invariance under translation is trivially satisfied by using relative positions as inputs.

key atomic interactions. Nevertheless, the core operation in these works remains the convolution filter based on the CG tensor product. In contrast, GotenNet (Aykent & Xia, 2025) proposes an effective tensor-product-free structure that combines edge attention with hierarchical refinement mechanisms, achieving SOTA performance in certain molecular benchmarks. In summary, the requirement of equivariance typically constrains model architectures to specific, rigid equivariant operations. Our work transforms the steerable vector into the spherical spatial domain. This transformation enables the application of more flexible and highly non-linear structures within message passing. Furthermore, we introduce an attention mechanism to weight the compositions of embeddings directly in the spatial domain, a method fundamentally different from previous approaches that only used edge attention.

**Spherical Fourier Transform**  Any square-integrable function $f(\vec{\mathbf{p}})$ defined over the sphere $\mathbb{S}^2$ can be expressed in a spherical harmonic basis via the spherical Fourier transform $\mathcal{F}$:

$$f(\vec{\mathbf{p}}) = \mathcal{F}(\mathbf{x}) = \sum_{l=0}^{\infty} \sum_{m=-l}^{l} x^{(l,m)} Y^{(l,m)}(\vec{\mathbf{p}}), \tag{4}$$

where $x^{(l,m)}$ is the spherical Fourier coefficient. Conversely, the coefficient can be obtained from the function $f(\vec{\mathbf{p}})$ via the inverse transform $\mathcal{F}^{-1}$:

$$x^{(l,m)} = \mathcal{F}^{-1}(f(\vec{\mathbf{p}})) = \int_{\Omega} f(\vec{\mathbf{p}}) Y^{(l,m)*}(\vec{\mathbf{p}}) d\Omega, \tag{5}$$

where $d\Omega$ denotes the unit sphere and $Y^{(l,m)*}(\cdot)$ is the conjugate of the spherical harmonic function. The function $f(\vec{\mathbf{p}})$ and its coefficients $\mathbf{x}$ are referred to as the representations in the spherical spatial domain and the harmonic domain, respectively. These processes are illustrated in Figure 1a. SCN (Zitnick et al., 2022) introduced an approach that applies a simple point-wise neural network in the spherical spatial domain. This technique has been subsequently adopted by several works (Passaro & Zitnick, 2023; Liao et al., 2024b) within their message-passing frameworks. However, point-wise neural networks inherently limit their expressiveness. Our work aims to extend the capabilities of learning within the whole spherical spatial domain.

## 3 MODEL

In this section, we first make a review of tensor product to detail our motivation. Then, we introduction the formulation of our EST and clarify its properties.

### 3.1 REVIEW OF CLEBSCH-GORDAN TENSOR PRODUCT

A common approach in equivariant models is to incorporate steerable vectors with multiple degrees for node or edge features and then employing CG tensor products $\otimes : \mathbb{V}_{l_1} \times \mathbb{V}_{l_2} \to \mathbb{V}_l$, which is defined as:

$$(\mathbf{x}_1 \otimes \mathbf{x}_2)^{(l,m)} = \sum_{m_1=-l_1}^{l_1} \sum_{m_2=-l_2}^{l_2} \mathcal{C}_{(l_1,m_1)(l_2,m_2)}^{(l,m)} \mathbf{x}_1^{(l_1,m_1)} \mathbf{x}_2^{(l_2,m_2)}, \tag{6}$$

where $\mathcal{C}_{(l_1,m_1)(l_2,m_2)}^{(l,m)}$ are the CG coefficients, a sparse tensor yielding non-zero terms when $|l_1 - l_2| \leq l \leq (l_1 + l_2)$. These products enable interactions between various combinations of $l_1$ and $l_2$, crucial for the expressive power of MPNNs in representing latent equivariant functions (Dym & Maron, 2021; Joshi et al., 2023). However, CG tensor products lack high non-linearities (Brandstetter et al., 2022), often requiring the stacking of multiple layers to capture complex molecular features. Furthermore, their high computational complexity ($\mathcal{O}(L^6)$) (Luo et al., 2024) makes architectures with stacked tensor product layers computationally expensive (Brandstetter et al., 2022). While techniques like Gate mechanisms (Brandstetter et al., 2022; Liao & Smidt, 2023) and geometry-aware tensor attention (Aykent & Xia, 2025) can be used mitigate this limitation, they apply non-linear transformations to the invariant ($l = 0$) representations. Higher-degree representations ($l > 0$) are then updated through a unified scaling multiplication:

$$C_{inv} = \mathrm{NL}(\mathbf{x}^{(0)}, \sum_{l=1}^{L} \mathrm{d}(\mathbf{x}^{(l)})), \quad \text{and} \quad \tilde{\mathbf{x}}^{(l)} = C_{inv} \cdot \mathbf{x}^{(l)}, \tag{7}$$

where $C_{inv}$ is an invariant scalar, $\mathrm{NL}(\cdot)$ is a non-linear network, and $\mathrm{d}(\cdot) : \mathbb{V}_l \to \mathbb{V}_0$ maps equivariant features to invariants, such as an inner product. This indirect exchange of information between higher-degree ($l > 0$) representations still limits the overall expressiveness.

### 3.2 EQUIVARIANT SPHERICAL TRANSFORMER

#### 3.2.1 NODE EMBEDDING

**Steerable Representation**   For representation with steerable vectors, we first define a maximal degree $L$. Each node $n$'s embedding $\mathbf{x}_n$ comprises $C$ channels, and each channel $c$ is a concatenation of steerable vectors from degree 0 to $L$: $\mathbf{x}_{n,c}^{(0 \rightarrow L)} = [x_{n,c}^{(0)}, \mathbf{x}_{n,c}^{(1)}, ..., \mathbf{x}_{n,c}^{(L)}]$. As a result, the dimension of a steerable node embedding is $(L+1)^2 \times C$.

**Spatial Representation**   Steerable representations $\mathbf{x}_{n,c}$ can be transformed into a spherical function $f_{n,c}(\vec{\mathbf{p}})$ via equation 4, with the summation truncated at a maximum degree $L$. Furthermore, $f_{n,c}(\vec{\mathbf{p}})$ is discretely represented by sampling $S$ points on the sphere, yielding a spatial node embedding $\mathbf{f}_n^* \in \mathbb{R}^{S \times C}$. It can be expressed as the concatenation over sampled points:

$$\mathbf{f}_{n,c}^* = [(\mathbf{x}_{n,c}^{(0 \rightarrow L)})^T \mathbf{Y}^{(0 \rightarrow L)}(\vec{\mathbf{p}}_1), (\mathbf{x}_{n,c}^{(0 \rightarrow L)})^T \mathbf{Y}^{(0 \rightarrow L)}(\vec{\mathbf{p}}_2), ..., (\mathbf{x}_{n,c}^{(0 \rightarrow L)})^T \mathbf{Y}^{(0 \rightarrow L)}(\vec{\mathbf{p}}_S)]. \quad (8)$$

We use the term $\mathbf{f}_{n,c,s}^*$ to denote the signal at the sampled point $\vec{\mathbf{p}}_s$ of channel $c$ of node $n$. The inverse transform in equation 5 can convert $\mathbf{f}_n^*$ back to $\mathbf{x}_n$.

The steerable and spatial representations can be converted into each other. This property allows our proposed EST to be easily integrated as a plug-and-play module into a variety of equivariant models.

**The significance of operating in the spatial domain**   In the spatial representation, the explicit degree and order are no longer directly accessible. Instead, higher-degree information is encoded as geometric features distributed across the spherical surface, as illustrated in Figure 1(a). Intuitively, by modeling the interdependencies of these geometric features, the model can effectively exchange information between implicit degrees, thereby approximating the capability of the tensor product discussed in Section 3.1. However, prior methods (Zitnick et al., 2022; Passaro & Zitnick, 2023; Liao et al., 2024b) only consider the feature dependencies across channels within the same sampling point $\vec{\mathbf{p}}_s$ (e.g. $[\mathbf{f}_{c=1,s}^*, \mathbf{f}_{c=2,s}^*, ..., \mathbf{f}_{c=C,s}^*]$), as depicted in Figure 1(c), overlooking geometric features composed of multiple points. In contrast, our proposed EST can model dependencies across various sampling points (e.g. $[\mathbf{f}_{s=1}^*, \mathbf{f}_{s=2}^*, ...\mathbf{f}_{s=S}^*]$). We make two key claims: (i) our EST is SO(3)-equivariant by an uniform FL sampling strategy on the sphere, and (ii) its expressive power can surpass that of tensor product-based frameworks.

#### 3.2.2 SPHERICAL ATTENTION

To prepare the data, we flatten all sampled points from $\mathbf{f}^*$ into a sequential format, where each point $s$ represented by a $C$-dimensional feature vector. We then apply an attention mechanism (Vaswani et al., 2017):

$$a_{s=i,s=j} = \frac{\exp(\mathbf{Q}_{s=i}\mathbf{K}_{s=j}^T/\sqrt{C})}{\sum_{k=1}^S \exp(\mathbf{Q}_{s=i}\mathbf{K}_{s=k}^T/\sqrt{C})}, \quad \text{and} \quad \tilde{\mathbf{f}}_{s=i}^* = \sum_{j=1}^S a_{s=i,s=j}\mathbf{V}_{s=j}, \quad (9)$$

where the $\mathbf{Q}$, $\mathbf{K}$, and $\mathbf{V}$ matrices are obtained through point-wise linear transformations of $\mathbf{f}_{s=i}^*$. The spherical attention is the core of EST. We also incorporate the pre-LayerNorm strategy (Xiong et al., 2020) and feedforward networks (FFN) in EST. Our EST is applied in two ways: (i) for interactions between two spatial representations, $\mathbf{Q}$ is derived from one, and $\mathbf{K}$ and $\mathbf{V}$ from the other; (ii) for learning features within a single spatial representation, $\mathbf{Q}$, $\mathbf{K}$, and $\mathbf{V}$ are all derived from it.

**Equivariance and sampling strategies**   We demonstrate that the equivariance of the EST module is contingent upon the uniformity of the sampling points used in the FT. We begin with the following theorem:

**Theorem 1.** *Let $\mathcal{P} = \{\vec{\mathbf{p}}_1, \ldots, \vec{\mathbf{p}}_N\}$ be a set of sampled points used for the FT. The EST module is SO(3)-equivariant if the set $\mathcal{P}$ satisfies the ideally rotational closure property under $SO(3)$. That is, for any point $\vec{\mathbf{p}}_i \in \mathcal{P}$ and any rotation $\mathbf{R} \in SO(3)$, there exists a point $\vec{\mathbf{p}}_j \in \mathcal{P}$ such that $\vec{\mathbf{p}}_j = \mathbf{R}\vec{\mathbf{p}}_i$. In this case, the equivariance of EST theoretically holds:*

$$\mathcal{F}^{-1}\big(\text{EST}\big(\mathcal{F}(\mathbf{D}\mathbf{x})\big)\big) = \mathbf{D}\mathcal{F}^{-1}\big(\text{EST}\big(\mathcal{F}(\mathbf{x})\big)\big), \quad (10)$$

*where $\mathbf{D}$ is an arbitrary Wigner-D rotation matrix.*

The proof is provided in Appendix C.1. As demonstrated by Theorem 1, $SO(3)$-equivariance relies on the rotational closure of the sampled point set $\mathcal{P}$. **Ideal closure requires infinite sampling points, a condition that is violated by the finite set $\mathcal{P}$ used in practice.** To mitigate this limitation, we utilize a uniform sampling strategy such that the closure relationship, $\vec{\mathbf{p}}_j = \mathbf{R}\vec{\mathbf{p}}_i$, is satisfied for a dense subset of rotations $\mathbf{R} \in SO(3)$. In contrast, most prior works (Passaro & Zitnick, 2023; Liao

et al., 2024b) implement spherical FT using the e3nn library (Geiger et al., 2022), which results in point densities that vary drastically with the polar angle $\theta$ (see Figure 2(a) for visualization). To address this, we redefine the sampling implementation using Fibonacci Lattices (FL) (González, 2010). The Cartesian coordinate of each sampling point $s$ is defined by

$$\vec{\mathbf{p}}_s = [\sqrt{1 - z_s^2}\cos(2s\pi/\lambda), \sqrt{1 - z_s^2}\sin(2s\pi/\lambda), z_s], \tag{11}$$

where $z_s = \frac{2s-1}{S} - 1$ and $\lambda = \frac{1+\sqrt{5}}{2}$ denotes the golden ratio. The equation 11 ensures the sampled points are distributed on the spherical surface, and different regions have similar densities (see Figure 2(b)). While FL sampling provides a high degree of uniformity, we further enhance this by simulating the points' motion on the sphere through consistent repulsions. This process, which resembles a simple molecular dynamics simulation, where each atom-like point is confined to the spherical surface, with forces determined by the distances between them.

The orthogonality of the basis functions in equation 4 is defined by $\int_\Omega Y^{(l,m)}(\vec{\mathbf{p}})Y^{(l',m')*}(\vec{\mathbf{p}})d\Omega = \delta_{ll'}\delta_{mm'}$, where $\delta_{ij}$ is the Kronecker delta. This work utilizes real-valued spherical harmonics, which possess the property of being self-conjugate: $Y^{(l,m)*}(\vec{\mathbf{p}}) = Y^{(l,m)}(\vec{\mathbf{p}})$. Then we transform equation 5 to a discrete inverse FT:

$$x^{l,m} = \sum_{s=1}^{S} \lambda^{(l,m)}\mathbf{f}_s^* Y^{(l,m)*}(\vec{\mathbf{p}}_s). \tag{12}$$

To ensure orthogonality in the code implementation, we adopt the simple sum of squares rule to define the quadrature weights:

$$\lambda^{(l,m)} = \frac{1}{\sum_{s=1}^{S} |Y^{(l,m)}(\vec{\mathbf{p}}_s)|^2}. \tag{13}$$

A detailed discussion comparing our sampling method with the e3nn implementation is provided in Appendix C.1. Furthermore, our ablation experiments (Table 6) demonstrate that sampling methods with poor uniformity can severely compromise equivariance.

**Expressiveness**  As demonstrated by Dym & Maron (2021); Joshi et al. (2023); Cen et al. (2024), the expressive power of CG tensor product-based frameworks is constrained by the maximum degree $L$, a finding corroborated by ablation experiments (Table 5). In contrast, our proposed EST module can project steerable representations of any degree onto the sphere and perform flexible, learnable operations. This allows the resulting features to be reprojected back to any desired degree, thereby potentially overcoming the expressiveness limitations of tensor products.

**Proposition 2.** *Under the assumption of no information loss in the spherical Fourier transform and its inverse, the function space spanned by the spherical Transformer tends to be a superset of that spanned by the CG tensor product.*

The detail is provided in Appendix C.2. We note that a spherical FT with no information loss is attainable by satisfying the Nyquist sampling rate, which requires the number of sampling points on the sphere satisfies $S \geq (2L)^2$ (Zitnick et al., 2022), where the term $L$ is the maximal degree of steerable representation. Thus, Proposition 2 demonstrates the potential for substituting tensor products with EST. An empirical comparison of the expressiveness of EST and tensor product-based operations is presented in Section 4.3, where our EST significantly surpasses the upper bound of tensor products.

**Relative Orientation Embedding**  In its basic form, spherical attention determines the dependencies between points based solely on their feature values, ignoring their relative positions in the input sequence. This means the output feature of a given point remains unchanged even if the order of all other points is altered. Such behavior is undesirable, as different point orderings correspond to distinct spherical functions. To enable the model to distinguish between these configurations, we introduce a relative orientation embedding by augmenting the query ($\mathbf{Q}$) and key ($\mathbf{K}$) vectors with orientation information:

$$\mathbf{Q}_{s=i} := [\mathbf{Q}_{s=i}, \vec{\mathbf{p}}_{s=i}], \quad \mathbf{K}_{s=j} := [\mathbf{K}_{s=j}, \vec{\mathbf{p}}_{s=j}], \quad \forall i, j \in \{1, 2, ..., S\}. \tag{14}$$

Consequently, the inner product between $\mathbf{Q}_{s_i}$ and $\mathbf{K}_{s_j}$ now incorporates a term related to the orientation-aware inner product $\vec{\mathbf{p}}_{s_i}^T\vec{\mathbf{p}}_{s_j}$. Crucially, this augmentation does not compromise the equivariant inductive bias of EST, as proven in Appendix C.3. Our approach shares conceptual similarities with Rotary Position Embeddings (RoPE) in NLP (Su et al., 2024), where the relative

position embedding is invariant under transformation. However, our method fundamentally differs from RoPE by addressing rotation invariance instead of permutation invariance.

### 3.2.3 MIXTURE OF HYBRID EXPERTS

To enhance the model's representational capacity, the FFNs in EST is reformed within a Mixture of Experts (MoE) framework (Shazeer et al., 2017). Starting with the steerable representation $\mathbf{x}$, the invariant ($l = 0$) component is used to compute expert weights through a gate function:

$$G(\mathbf{x}) = \text{Softmax}(\mathbf{x}^{(0)}\mathbf{W}_G), \tag{15}$$

where $\mathbf{W}_G \in \mathbb{R}^{C \times E}$ is a trainable weight matrix. We then employ two distinct expert structures: (1) a steerable FFN, operating on steerable inputs and consisting of two degree-wise linear layers with an intermediate Gate activation (Liao & Smidt, 2023); and (2) a spherical FFN, processing spatial inputs and comprising two point-wise linear layers with an intermediate SiLU activation (Elfwing et al., 2018). The final output is a weighted combination of the outputs from these hybrid experts, with the weights determined by equation 15. These two expert types offer a crucial trade-off: the steerable FFN guarantees strict equivariance, while the spherical FFN provides enhanced expressive power. We therefore adapt their numbers based on the specific demands of the task. Further details and ablation studies can be found in Appendix D.

### 3.3 APPLYING THE EST MODULE TO EQUIVARIANT MODELS

We can apply EST module into equivariant models by two primary ways:

- **Message block:** We replace the conventional message block, often a tensor product like $\text{Combine}(\mathbf{x}_i, \mathbf{x}_j) \otimes \vec{r}_{ij}$, with a EST. Here, the query ($\mathbf{Q}$) is formed from a combination of the embeddings of nodes $i$ and $j$, while the key ($\mathbf{K}$) and value ($\mathbf{V}$) are derived from the spherical harmonic representation of the relative position vector $\vec{r}_{ij}$.
- **Update block:** To replace update block, all three attention components—query ($\mathbf{Q}$), key ($\mathbf{K}$), and value ($\mathbf{V}$)—are derived directly from the aggregated messages.

We provide an architecture in Appendix D.4 that uses the strategies mentioned above. Moreover, our approach leverages EST as a plug-and-play module that integrates seamlessly into existing equivariant models. Specifically, **we recommend replacing the standard update blocks—which require strong expressiveness to learn richer representations from complex aggregated message—with our EST-based update blocks.** This allows EST-based architectures to benefit from the computational efficiency and stability of existing message blocks while still enhancing their expressive power.

## 4 EXPERIMENTS

In this section, we construct experiments to investigate the efficacy of the proposed method. We evaluate its performance on the S2EF/IS2RE tasks from the OC20 (Chanussot et al., 2021) benchmark and QM9 (Ramakrishnan et al., 2014) tasks. Our analysis involves a comparison with a wide range of baseline models, including Schnet (Schütt et al., 2018), PaiNN (Schütt et al., 2021), SEGNN (Brandstetter et al., 2022), TFN (Thomas et al., 2018), Dimenet++ (Klicpera et al., 2020), Equiformer (Liao & Smidt, 2023), Equiformerv2 (Liao et al., 2024b) and EScAIP (Qu & Krishnapriyan, 2024). Specifically for the OC20 benchmark, we extend our comparison to include SpinConv (Shuaibi et al., 2021), GemNet (Gasteiger et al., 2021; 2022), SphereNet (Liu et al., 2021), SCN (Zitnick et al., 2022) and eSCN (Passaro & Zitnick, 2023). Similarly, our QM9 experiments are augmented with comparisons to TorchMD-NET (Thölke & Fabritiis, 2022), EQGAT (Le et al., 2022) and GotenNet (Aykent & Xia, 2025). The specific configurations employed for each baseline model can be found in Appendix E.1.

### 4.1 OC20 RESULTS

**Dataset and Configurations** The OC20 dataset, comprising over 130 million molecular structures with force and energy labels, covers a broad spectrum of materials, surfaces, and adsorbates. We evaluate our models on two core sub-datasets of OC20: S2EF and IS2RE. All experimental configurations are detailed in Appendix E.2. Notably, We did not use very deep models or very long training schedules. For instance, our S2EF experiments utilize a 8-layer architecture, and IS2RE experiments employ a 6-layer one, significantly fewer than many advanced methods. Furthermore, as mentioned in Section 3.3, to enhance computational efficiency and facilitate a more direct comparison of different fundamental modules, we replace the EST architecture's message module with the graph attention modules from EquiformerV2 and Equiformer for S2EF and IS2RE tasks, respectively.

Table 1: S2EF validation results. $\lambda_E$ is the coefficient of the energy loss. "$\tau$" denotes the test results.

| Dataset | Model | Number of parameters | Throughput Samples/s | Energy MAE (meV) ↓ | Force MAE (meV/Å) ↓ |
|---|---|---|---|---|---|
| All+MD | EScAIP-Small | 83M | - | **229** | 15.1 |
| | EquiformerV2 ($\lambda_E = 4$, 8 layers) | 31M | 7.1 | 232 | 16.3 |
| All | SchNet | 9.1M | - | 549 | 56.8 |
| | DimeNet++-L-F+E | 10.7M | 4.6 | 515 | 32.8 |
| | SpinConv | 8.5M | 6.0 | 371 | 41.2 |
| | GemNet-dT | 32M | 25.8 | 315 | 27.2 |
| | GemNet-OC | 39M | 18.3 | 244 | 21.7 |
| | SCN$^\tau$ | 271M | - | 244 | 17.7 |
| | eSCN$^\tau$ | 200M | 2.9 | 242 | 17.1 |
| | EquiformerV2 ($\lambda_E = 2$, 20 layers) | 153M | 1.8 | 236 | 15.7 |
| | EquiformerV2 + EST Update ($\lambda_E = 4$, 8 layers) | 45M | 6.8 | **229** | 15.8 |
| | EquiformerV2 + EST Update ($\lambda_E = 2$, 8 layers) | 45M | 6.8 | 232 | **15.0** |

**S2EF Results** We trained an 8-layer EST model on the S2EF-All dataset. Each validation set was divided into four similarly-sized subsets: In-Domain (ID), Out-of-Domain Catalysis (OOD Cat), Out-of-Domain Both (OOD Both). Consistent with prior work (Passaro & Zitnick, 2023; Liao et al., 2024b), the best model was selected based on its performance on the ID validation subset during training and subsequently evaluated across all validation subsets. For comparison with SCN and eSCN, we report their test results due to the absence of publicly available validation results; previous work (Liao et al., 2024b) indicates that the validation and test sets have a similar distribution and the test results are usually slightly better than the validation results. We compared our results with two variants of the EquiformerV2 model: a deep 20-layer architecture and one trained with additional MD dataset. As shown in Table 1, EST outperformed both EquiformerV2 variants in energy and forces prediction. Furthermore, EST achieved competitive results in force prediction, surpassing several bigger baselines including EScAIP, SCN and eSCN.

Table 2: IS2RE results of models trained on IS2RE training dataset. Bold and underline indicate the best result, and the second best result, respectively.

| Model | Energy MAE (meV) ↓ | | | | EwT (%) ↑ | | | |
|---|---|---|---|---|---|---|---|---|
| | ID | OOD Ads | OOD Cat | OOD Both | ID | OOD Ads | OOD Cat | OOD Both |
| SchNet | 639 | 734 | 662 | 704 | 2.96 | 2.33 | 2.94 | 2.21 |
| PaiNN | 575 | 783 | 604 | 743 | 3.46 | 1.97 | 3.46 | 2.28 |
| TFN | 584 | 766 | 636 | 700 | 4.32 | 2.51 | 4.55 | 2.66 |
| DimeNet++ | 562 | 725 | 576 | 661 | 4.25 | 2.07 | 4.10 | 2.41 |
| GemNet-dT | 527 | 758 | 549 | 702 | 4.59 | 2.09 | 4.47 | 2.28 |
| GemNet-OC | 560 | 711 | 576 | 671 | 4.15 | 2.29 | 3.85 | 2.28 |
| SphereNet | 563 | 703 | 571 | 638 | 4.47 | 2.29 | 4.09 | 2.41 |
| SEGNN | 533 | 692 | 537 | 679 | **5.37** | 2.46 | 4.91 | 2.63 |
| SCN | 516 | **643** | 530 | 604 | 4.92 | **2.71** | 4.42 | **2.76** |
| Equiformer | 504 | 688 | 521 | 630 | 5.14 | 2.41 | 4.67 | 2.69 |
| + EST Update | **501** | 652 | **502** | **578** | 5.16 | 2.67 | **5.16** | **2.76** |

**IS2RE Results** We trained our model directly on the IS2RE training set for energy prediction, without utilizing S2EF data. Related results, including the Mean Absolute Error (MAE) for energy and the percentage of Energies Within a Threshold (EwT) for each model, are summarized in Table 2. EST achieves SOTA performance on ID, OOD Cat, OOD Both and the second best on OOD Ads. It is notable that the current SOTA model on OOD Ads, SCN, employs complex 16-layer architecture with over 100M parameters while EST achieves closely competitive performance using a unified 6-layer structure comprising just 32.47 million parameters. Crucially, SCN relax equivariance to gain expressiveness, which can potentially lead to unstable predictions under input rotation. EST, conversely, provides more stable predictions owing to its stronger equivariance. Furthermore, in these IS2RE experiments, the EST architecture incorporates the message block from Equiformer. The key architectural difference resides in the update block. We observe that EST consistently surpasses the Equiformer model across all evaluated metrics, providing further evidence for the effectiveness of the proposed EST module and the model merging strategy discussed in Section 3.3.

### 4.2 QM9 RESULTS

**Dataset and Configurations** The QM9 benchmark comprises quantum chemical properties for a relevant, consistent, and comprehensive chemical space of 134k equilibrium small organic molecules

Table 3: Results on QM9 dataset for various properties. † denotes using different data partitions. Gray indicates the best results within different comparison modes, and bold indicates the best results among all models.

| Task Units | $\alpha$ $bohr^3$ | $\Delta\varepsilon$ meV | $\varepsilon_{HOMO}$ meV | $\varepsilon_{LUMO}$ meV | $\mu$ D | $C_v$ cal/(mol K) | $G$ meV | $H$ meV | $R^2$ $bohr^3$ | $U$ meV | $U_0$ meV | $ZPVE$ meV |
|---|---|---|---|---|---|---|---|---|---|---|---|---|
| SchNet | .235 | 63 | 41 | 34 | .033 | .033 | 14 | 14 | .073 | 19 | 14 | 1.70 |
| TFN† | .223 | 58 | 40 | 38 | .064 | .101 | - | - | - | - | - | - |
| DimeNet++ | .044 | 33 | 25 | 20 | .030 | .023 | 8 | 7 | .331 | 6 | 6 | 1.21 |
| PaiNN | .045 | 46 | 28 | 20 | .012 | .024 | 7.35 | 5.98 | .066 | 5.83 | 5.85 | 1.28 |
| TorchMD-NET | .059 | 36 | 20 | 18 | .011 | .026 | 7.62 | 6.16 | .033 | 6.38 | 6.15 | 1.84 |
| SEGNN† | .060 | 42 | 24 | 21 | .023 | .031 | 15 | 16 | .660 | 13 | 15 | 1.62 |
| EQGAT | .053 | 32 | 20 | 16 | .011 | .024 | 23 | 24 | .382 | 25 | 25 | 2.00 |
| EquiformerV2 | .050 | 29 | 14 | 13 | .010 | .023 | 7.57 | 6.22 | .186 | 6.49 | 6.17 | 1.47 |
| EST$^E$ | .042 | 28 | 13.4 | 12.4 | .0113 | .022 | 7.03 | 5.94 | .298 | 5.92 | 5.64 | 1.31 |
| EST$^G$ | .040 | 26 | 13.2 | 12.3 | .0101 | .021 | 7.05 | 5.81 | .229 | 5.85 | 5.57 | 1.27 |
| Equiformer | .046 | 30 | 15 | 14 | .011 | .023 | 7.63 | 6.63 | .251 | 6.74 | 6.59 | 1.26 |
| + EST Update | .041 | 29 | 14 | 13 | .011 | .021 | 7.18 | 6.17 | .227 | 6.35 | 6.32 | 1.27 |
| GotenNet-S | .033 | 21.2 | 16.9 | 13.9 | .0075 | .020 | 5.50 | 3.70 | .029 | 3.67 | 3.71 | 1.09 |
| + EST Update | .030 | 31.2 | 16.7 | 13.3 | .0070 | **.019** | 5.40 | 3.71 | .029 | 3.70 | 3.70 | 1.19 |
| GotenNet-B | .032 | 20.5 | 15.2 | 13.0 | .0072 | **.019** | 5.19 | 3.44 | .027 | 3.49 | 3.43 | 1.09 |
| + EST Update | .030 | 31.6 | 15.0 | 13.1 | .0071 | **.019** | 5.13 | 3.60 | .028 | 3.58 | 3.55 | 1.17 |
| GotenNet-L | **.028** | **19.8** | **13.4** | **12.2** | **.0067** | **.019** | **4.98** | **3.30** | **.024** | **3.41** | **3.37** | **1.08** |
| + EST Update | **.028** | 30.8 | 13.7 | 12.8 | .0070 | **.019** | 5.26 | 3.41 | .027 | 3.50 | 3.53 | 1.18 |

containing up to 29 atoms. Each atom is represented by its 3D coordinates and an embedding of its atomic type (H, C, N, O, F). We developed three model architectures: the first model, denoted as EST$^E$ or EST$^G$, employs the EST-based message-passing shown in Figure 4. The second model, denoted as Equiformer+EST, integrates Equiformer's Graph Attention (GA) message block with EST's update block. The third model, denoted as GotenNet+EST, integrates GotenNet's geometry-aware tensor attention (GATA) message block with EST's update block. For additional configuration specifics, please refer to Appendix E.2.

**Results** Given that the QM9 dataset is considerably smaller than OC20, models are more prone to overfitting if equivariance is destroyed, which means the experiments on QM9 have higher requirements for the model's equivariance. However, discrete sampling in EST may lead to minor deviations from perfect equivariance (Table 6). Conversely, the comparison methods in Table 3, except for EquiformerV2, are all strictly equivariant methods.

In the first comparison, we compare the architecture fully based on EST with various molecular models. For fairness, we train EST with a training strategy similar to Equiformer (EST$^E$) and GotenNet (EST$^G$). From Table 3, EST achieves the best performance on nine out of the twelve tasks, except for GotenNet. However, we also find that even with the training strategy of GotenNet, EST still lags behind the GotenNet model. We argue that this is because the architecture that uses spherical attention in both the message block and the update block exacerbates the lack of equivariance, thereby limiting the model's ability to approach the saturated performance in QM9 tasks.

In our second comparison, we integrate EST as a plug-in into Equiformer and GotenNet to assess its capability as a plug-and-play module. In the Equiformer experiment, we observe performance improvements across all properties except for $ZPVE$. Using a configuration of 6 steerable experts and 4 spherical experts within EST, the model's throughput drops from 480 to 302 samples/second. This acceptable drop demonstrates the computational scalability of EST. As detailed in Section 3.3, the strategy of inserting EST into the update block is designed to process complex aggregated messages with less overall computation. This allows us to effectively balance the enhancement of model expressiveness with the control of computational efficiency. Further experiments exploring various expert combinations are available in Appendix E.

In the GotenNet+EST experiments, we consistently employed an update block with 1 steerable expert and 1 spherical expert across all models. The results presented in Table 3 yield two principal observations: 1. EST does not yield stable improvements on GotenNet. Its enhancements are predominantly concentrated on non-energy properties such as $\alpha$, $\mu$ and $\varepsilon_{HOMO}$. Conversely, it exhibits poor performance on core energy-related properties ($G, H, U$, and $U_0$). 2. The improvements provided by EST weaken as the depth of GotenNet increases. In GotenNet-S, EST improves performance in 6 tasks, but in the deeper GotenNet-L, EST achieves no improvement. We hypothesize that this behavior stems from two factors. First, GotenNet already achieves excellent performance nearing the saturation limit for the QM9 dataset, whose labels are derived from inherently noisy

quantum chemical simulations. Second, the loss of equivariance introduced by EST may accumulate across an increasing number of layers, degrading the model's generalization performance on QM9.

**Computational Efficiency**   We evaluated the train and inference latency of EST-integrated models on a single Nvidia H800. As shown Table 4, EST adds inference latency of $< 20\%$ on GotenNet-S and GotenNet-L, respectively. The addition is higher for Equiformer ($21\%$) due to a greater number of experts.

Table 4: Comparison of train and inference latency on the QM9 dataset. The bacth sizes of all models are set to 32.

| Model | EST | Equiformer | Equiformer+EST | GotenNet-S | GotenNet-S+EST | GotenNet-L | GotenNet-L+EST |
|---|---|---|---|---|---|---|---|
| Train (Samples/s) | 286 | 185 | 264 | 55 | 64 | 116 | 130 |
| Inference (Samples/s) | 128 | 67 | 81 | 35 | 41 | 106 | 115 |

## 4.3 ABLATION STUDY

In this section, we explore two key properties of EST: expressiveness and equivariance. The other experiments can be found in Appendix E, where we comprehensively investigate the influence of building components, including SA, spherical relative orientation embedding and hybrid experts.

Table 5: Experiments on Rotationally symmetric structures.

| GNN Layer | 2 fold | 3 fold | 10 fold | 100 fold | GNN Layer | 2 fold | 3 fold | 5 fold | 10 fold |
|---|---|---|---|---|---|---|---|---|---|
| $EST_{L=1}$ | $100.0 \pm 0.0$ | $100.0 \pm 0.0$ | $100.0 \pm 0.0$ | $100.0 \pm 0.0$ | $TFN/MACE_{L=1}$ | $50.0 \pm 0.0$ | $50.0 \pm 0.0$ | $50.0 \pm 0.0$ | $50.0 \pm 0.0$ |
| $EST_{L=2}$ | $100.0 \pm 0.0$ | $100.0 \pm 0.0$ | $100.0 \pm 0.0$ | $100.0 \pm 0.0$ | $TFN/MACE_{L=2}$ | $100.0 \pm 0.0$ | $50.0 \pm 0.0$ | $50.0 \pm 0.0$ | $50.0 \pm 0.0$ |
| $EST_{L=3}$ | $100.0 \pm 0.0$ | $100.0 \pm 0.0$ | $100.0 \pm 0.0$ | $100.0 \pm 0.0$ | $TFN/MACE_{L=3}$ | $100.0 \pm 0.0$ | $100.0 \pm 0.0$ | $50.0 \pm 0.0$ | $50.0 \pm 0.0$ |
| $EST_{L=5}$ | $100.0 \pm 0.0$ | $100.0 \pm 0.0$ | $100.0 \pm 0.0$ | $100.0 \pm 0.0$ | $TFN/MACE_{L=5}$ | $100.0 \pm 0.0$ | $100.0 \pm 0.0$ | $100.0 \pm 0.0$ | $50.0 \pm 0.0$ |
| $EST_{L=10}$ | $100.0 \pm 0.0$ | $100.0 \pm 0.0$ | $100.0 \pm 0.0$ | $100.0 \pm 0.0$ | $TFN/MACE_{L=10}$ | $100.0 \pm 0.0$ | $100.0 \pm 0.0$ | $100.0 \pm 0.0$ | $100.0 \pm 0.0$ |

**Expressiveness**   Following (Joshi et al., 2023), we employ evaluation metrics that distinguish $n$-fold symmetric structures to precisely assess EST's expressive power. We first construct a single-layer message-passing layer based on Figure 4, consistent with tensor product-based operations (TFN, MACE). Additionally, we remove the steerable FFN for a clear comparison. As demonstrated in (Dym & Maron, 2021; Joshi et al., 2023), the expressive power of tensor product-based models is limited by the maximum degree $L$, failing to perfectly distinguish $n$-fold symmetric structures when $n > L$. From Table 5, we observe that EST significantly breaks through this theoretical boundary. It allows the model to distinguish symmetric structures even with very high fold (e.g., using 1-degree EST for 100-fold structures).

Table 6: Equivariance evaluation with different sampling strategies.

| Sampling Strategy/Number | 1-layer | 2-layer | 3-layer | 4-layer | 5-layer | 6-layer |
|---|---|---|---|---|---|---|
| FL / $S = 64$, w/o optim. | 0.0010 | 0.0011 | 0.0010 | 0.0018 | 0.0022 | 0.0024 |
| FL / $S = 64$, w optim. | 0.0002 | 0.0001 | 0.0003 | 0.0003 | 0.0002 | 0.0002 |
| FL / $S = 256$, w/o optim. | 0.0002 | 0.0002 | 0.0001 | 0.0002 | 0.0001 | 0.0003 |
| e3nn / $S = 210$, w/o optim. | 0.0084 | 0.1006 | 0.0366 | 0.0593 | 0.0046 | 0.0199 |

**Equivariance**   While we theoretically demonstrate that EST can achieve strict equivariance, perfect uniform sampling is challenging in practice, potentially leading to loss of equivariance. To investigate it, we build untrained networks with 1 to 6 layers based on a EST-only message passing, controlling the number of spherical sampling points and choosing whether to use molecular dynamics-like optimization. We randomly select 1000 molecules from QM9, compute their outputs $y_1, ..., y_{1000}$, and then compute the outputs after applying random rotations, $\hat{y}_1, ..., \hat{y}_{1000}$. The average absolute error $(1/1000) \sum_{i=1}^{1000} |y_i - \hat{y}_i|$ serves as our measure. As shown in Table 6, EST with FL sampling closely approximates strict equivariance even without training, and its equivariance further improves with an increasing number of sampling points or a dynamics-like optimization. Additionally, we evaluated the equivariance error of 6-layer trained EST models on 1000 data samples from the each task of QM9 dataset. Across all properties, the average error was $1.8 \times 10^{-5} \pm 0.7 \times 10^{-5}$.

## 5 CONCLUSION

We present EST, a new SE(3)-equivariant framework for modeling molecules. By integrating a Transformer structure with steerable vectors, EST offers greater expressive power than tensor-product-based frameworks. We showed EST's strong performance on the OC20 and QM9 datasets. A current limitation is the slight equivariance loss caused by spherical sampling. Future work will focus on improving spherical sampling uniformity. This could lead to better equivariance.

# REPRODUCIBILITY

Our datasets are all based on open-source datasets.The experimental methodology, data proportions, and hyperparameter settings are detailed in Appendix E.2. Our code and data are included in the anonymous repositories, which we mentioned in abstract.

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

APPENDIX

## A  THE USE OF LARGE LANGUAGE MODELS

In our work,we exclusively use LLMs for writing refinement, which means we first write a piece of text ourselves,and then use the LLM to correct grammar, formatting, andother issues. For our experiments, we only use LLM to fix bug.

## B  THE MATHEMATICS

### B.1  THE MATHEMATICS OF SPHERICAL HARMONICS

#### B.1.1  THE PROPERTIES OF SPHERICAL HARMONICS

The spherical harmonics $Y^{(l,m)}(\theta, \varphi)$ are the angular portion of the solution to Laplace's equation in spherical coordinates where azimuthal symmetry is not present. Some care must be taken in identifying the notational convention being used. In this entry, $\theta$ is taken as the polar (colatitudinal) coordinate with $\theta$ in $[0, \pi]$, and $\varphi$ as the azimuthal (longitudinal) coordinate with $\varphi$ in $[0, 2\pi)$.

Spherical harmonics satisfy the spherical harmonic differential equation, which is given by the angular part of Laplace's equation in spherical coordinates. If we define the solution of Laplace's equation as $F = \Phi(\varphi)\Theta(\theta)$, the equation can be transformed as:

$$\frac{\Phi(\varphi)}{\sin\theta}\frac{d}{d\theta}\left(\sin\theta\frac{d\Theta}{d\theta}\right) + \frac{\Theta(\theta)}{\sin^2\theta}\frac{d^2\Phi(\varphi)}{d\varphi^2} + l(l+1)\Theta(\theta)\Phi(\varphi) = 0 \qquad (16)$$

Here we omit the derivation process and just show the result. The (complex-value) spherical harmonics are defined by:

$$Y^{(l,m)}(\theta, \varphi) \equiv \sqrt{\frac{2l+1}{4\pi}\frac{(l-m)!}{(l+m)!}}P^{(l,m)}(\cos\theta)e^{im\varphi}, \tag{17}$$

where $P^{(l,m)}(\cos\theta)$ is an associated Legendre polynomial. Spherical harmonics are integral basis, which satisfy:

$$\begin{aligned}
&\int_0^{2\pi}\int_0^{\pi} Y^{(l_1,m_1)}(\theta,\varphi)Y^{(l_2,m_2)}(\theta,\varphi)Y^{(l_3,m_3)}(\theta,\varphi)\sin\theta d\theta d\varphi\\
&= \sqrt{\frac{(2l_1+1)(2l_2+1)(2l_3+1)}{4\pi}}\begin{pmatrix} l_1 & l_2 & l_3 \\ 0 & 0 & 0 \end{pmatrix}\begin{pmatrix} l_1 & l_2 & l_3 \\ m_1 & m_2 & m_3 \end{pmatrix},
\end{aligned} \tag{18}$$

where $\begin{pmatrix} l_1 & l_2 & l_3 \\ m_1 & m_2 & m_3 \end{pmatrix}$ is a Wigner 3j-symbol (which is related to the Clebsch-Gordan coefficients). We list a few spherical harmonics which are:

$$Y^{(0,0)}(\theta,\varphi) = \frac{1}{2}\sqrt{\frac{1}{\pi}},$$

$$Y^{(1,-1)}(\theta,\varphi) = \frac{1}{2}\sqrt{\frac{3}{2\pi}}\sin\theta e^{-i\varphi},$$

$$Y^{(1,0)}(\theta,\varphi) = \frac{1}{2}\sqrt{\frac{3}{\pi}}\cos\theta,$$

$$Y^{(1,1)}(\theta,\varphi) = \frac{-1}{2}\sqrt{\frac{3}{2\pi}}\sin\theta e^{i\varphi},$$

$$Y^{(2,-2)}(\theta,\varphi) = \frac{1}{4}\sqrt{\frac{15}{2\pi}}\sin^2\theta e^{-2i\varphi}, \tag{19}$$

$$Y^{(2,-1)}(\theta,\varphi) = \frac{1}{2}\sqrt{\frac{15}{2\pi}}\sin\theta\cos\theta e^{-i\varphi},$$

$$Y^{(2,0)}(\theta,\varphi) = \frac{1}{4}\sqrt{\frac{5}{\pi}}\left(3\cos^2\theta - 1\right),$$

$$Y^{(2,1)}(\theta,\varphi) = \frac{-1}{2}\sqrt{\frac{15}{2\pi}}\sin\theta\cos\theta e^{i\varphi},$$

$$Y^{(2,2)}(\theta,\varphi) = \frac{1}{4}\sqrt{\frac{15}{2\pi}}\sin^2\theta e^{2i\varphi},$$

In this work, we use the real-value spherical harmonics rather than the complex-value one, which can be written as :

$$Y^{0,0}(\theta, \varphi) = \sqrt{\frac{1}{4\pi}},$$

$$Y^{(1,-1)}(\theta, \varphi) = \sqrt{\frac{3}{4\pi}} \sin\varphi \sin\theta,$$

$$Y^{(1,0)}(\theta, \varphi) = \sqrt{\frac{3}{4\pi}} \cos\theta,$$

$$Y^{(1,1)}(\theta, \varphi) = \sqrt{\frac{3}{4\pi}} \cos\varphi \sin\theta,$$

$$Y^{(2,-2)}(\theta, \varphi) = \sqrt{\frac{15}{16\pi}} \sin(2\varphi) \sin^2\theta, \tag{20}$$

$$Y^{(2,-1)}(\theta, \varphi) = \sqrt{\frac{15}{4\pi}} \sin\varphi \sin\theta \cos\theta,$$

$$Y^{(2,0)}(\theta, \varphi) = \sqrt{\frac{5}{16\pi}} (3\cos^2\theta - 1),$$

$$Y^{(2,1)}(\theta, \varphi) = \sqrt{\frac{15}{4\pi}} \cos\varphi \sin\theta \cos\theta,$$

$$Y^{(2,2)}(\theta, \varphi) = \sqrt{\frac{15}{16\pi}} \cos(2\varphi) \sin^2\theta.$$

### B.1.2 Fourier transformation over $\mathbb{S}^2$

In the main paper, we show that any square-integrable function $f(\cdot)$ can thus be expanded as a linear combination of spherical harmonics:

$$f(\vec{\mathbf{p}}) = \sum_{l=0}^{\infty} \sum_{m=-l}^{l} \mathbf{x}^{(l,m)} Y^{(l,m)}(\vec{\mathbf{p}}), \tag{21}$$

where $\vec{\mathbf{p}} = (\theta, \varphi)$ denotes the orientations, like what we do in the main paper. The coefficient $\mathbf{x}^{(l,m)}$ can be obtained by the inverse transformation over $\mathbb{S}^2$, which is

$$\mathbf{x}^{l,m} = \int_{\Omega} f(\vec{\mathbf{p}}) Y^{(l,m)*}(\vec{\mathbf{p}}) d\Omega = \int_0^{2\pi} d\varphi \int_0^{\pi} d\theta \sin\theta f(\vec{\mathbf{p}}) Y^{(l,m)*}(\vec{\mathbf{p}}). \tag{22}$$

Using the fact $\mathbf{Y}^l(\mathbf{R}\vec{\mathbf{p}}) = \mathbf{D}^l(\mathbf{R})\mathbf{Y}^l(\vec{\mathbf{p}})$, and equation 21, we know

$$f(\mathbf{R}\vec{\mathbf{p}}) = \sum_{l=0}^{\infty} \sum_{m=-l}^{l} \mathbf{x}^{(l,m)} Y^{(l,m)}(\mathbf{R}\vec{\mathbf{p}}) = \sum_{l=0}^{\infty} \sum_{m=-l}^{l} \mathbf{x}^{(l,m)} \mathbf{D} Y^{(l,m)}(\vec{\mathbf{p}}). \tag{23}$$

Therefore, we can get the conclusion that spatial representation $f(\mathbf{R}\vec{\mathbf{P}})$ and $f(\vec{\mathbf{P}})$ is steerable, which can be represented by

$$f(\mathbf{R}\vec{\mathbf{p}}) = \mathbf{D}^{-1}\mathbf{x} = \mathbf{D}^T\mathbf{x}. \tag{24}$$

### B.1.3 The Relationship Between Spherical Harmonics and Wigner-D Matrix

One can also think of standard spherical harmonics as functions $\tilde{Y}_m^{(l)} : \mathrm{SO}(3) \to \mathbb{R}$ on $\mathrm{SO}(3)$ that are invariant to a sub-group of rotations via

$$Y^{(l,m)}(\mathbf{n}_{\alpha,\beta}) = Y^{(l,m)}(\mathbf{R}_{\alpha,\beta,\gamma}\mathbf{n}_x) =: \tilde{Y}^{(l,m)}(\mathbf{R}_{\alpha,\beta,\gamma}) .$$

Then, by definition, $\tilde{Y}(l,m)$ is invariant with respect to rotation angle $\gamma$, i.e.., $\forall_{\gamma \in [0,2\pi)}$ : $\tilde{Y}^{(l,m)}(\mathbf{R}_{\alpha,\beta,\gamma}) = \tilde{Y}^{(l,m)}(\mathbf{R}_{\alpha,\beta,0})$. This viewpoint of regarding the spherical harmonics as $\gamma$-invariant functions on $\mathrm{O}(3)$ helps us to draw the connection to the Wigner-D functions $D_{mn}^{(l)}$ that make up the $2l + 1 \times 2l + 1$ elements of the Wigner-D matrices. Namely, the $n = 0$ column of

Wigner-D functions are also $\gamma$-invariant and, in fact, correspond (up to a normalisation factor) to the spherical harmonics via

$$Y^{(l,m)}(\mathbf{n}_{\alpha,\beta}) = \frac{1}{\sqrt{2l+1}} D_{m0}^{(l)}(\mathbf{R}_{\alpha,\beta,\gamma}) \ . \tag{25}$$

### B.1.4 Equivariance of Clebsch-Gordan Tensor Product

The Clebsch-Gordan Tensor Product shows a strict equivariance for different group representations, which make the mixture representations transformed equivariant based on Wigner-D matrices. We use $\mathbf{D}^{l,m'm}$ to denote the element of Wigner-D matrix in the degree $l$. The Clebsch-Gordan coefficient satisfies:

$$\sum_{m_1',m_2'} \mathcal{C}_{(l_1,m_1')(l_2,m_2')}^{(l_0,m_0)} \mathbf{D}^{(l_1,m_1'm_1)}(g)\mathbf{D}^{(l_2,m_2'm_2)}(g)$$
$$= \sum_{m_0'} \mathbf{D}^{(l_0,m_0'm_0)}(g)\mathcal{C}_{(l_1,m_1)(l_2,m_2)}^{(l_0,m_0')} \tag{26}$$

Therefore, the spherical harmonics can be combined equivariantly by CG Tensor Product:

$$CG\left(\sum_{m_1'} \mathbf{D}^{(l_1,m_1m_1')}(g)Y^{(l_1,m_1')}, \sum_{m_2'} \mathbf{D}^{(l_2,m_2m_2')}(g)Y^{(l_2,m_2')}\right)$$
$$= \sum_{m_1,m_2} \mathcal{C}_{(l_1,m_1)(l_2,m_2)}^{(l_0,m_0)} \sum_{m_1'} \mathbf{D}^{(l_1,m_1m_1')}(g)Y^{(l_1,m_1')} \sum_{m_2'} \mathbf{D}^{(l_2,m_2m_2')}(g)Y^{(l_2,m_2')}$$
$$= \sum_{m_0'} \mathbf{D}^{(l_0,m_0m_0')}(g) \sum_{m_1,m_2} \mathcal{C}_{(l_1,m_1)(l_2,m_2)}^{(l_0m_0')} Y^{(l_1,m_1')}Y^{(l_2,m_2')} \tag{27}$$
$$= \sum_{m_0'} \mathbf{D}^{(l_0,m_0m_0')}(g)CG\left(Y^{(l_1,m_1')}, Y^{(l_2,m_2')}\right) \ .$$

Here, we omit the input argument of the spherical harmonics, which can represent any direction on the sphere. equation 27 represents a relationship between scalar. If we transform the scalar to vector or matrix like what we do in equation 3, equation 27 is equal to

$$(\mathbf{D}^{l_1}\mathbf{u} \otimes \mathbf{D}^{l_2}\mathbf{v})^{l_0} = \mathbf{D}^{l_0}(\mathbf{u} \otimes \mathbf{v})^{l_0}. \tag{28}$$

The tensor CG product mixes two representations to a new representation under special rule $|l_1 - l_2| \leq l \leq (l_1 + l_2)$. For example, 1.two type-0 vectors will only generate a type-0 representations; 2.type-$l_1$ and type-$l_2$ can generate type-$l_1 + l_2$ vector at most. Note that some widely-used products are related to tensor product: scalar product ($l_1 = 0, l_2 = 1, l = 1$), dot product ($l_1 = 1, l_2 = 1, l = 0$) and cross product ($l_1 = 1, l_2 = 1, l = 1$). However, for each element with $l > 0$, there are multi mathematical operation for the connection with weights. The relation between number of operations and degree is quadratic. Thus, as degree increases, the amount of computation increases significantly, making calculation of the CG tensor product slow for higher order irreps. This statement can be proven by the implementation of e3nn (o3.FullyConnectedTensorProduct).

### B.1.5 Learnable Parameters in Tensor Product

Previous works utilize the e3nn library (Geiger et al., 2022) to implement the corresponding tensor product. It is crucial to emphasize that the formulation of CG tensor product is devoid of any learnable parameters, as CG coefficients remain constant. In the context of e3nn, learnable parameters are introduced into each path, represented as $w(\mathbf{u}^{l_1} \otimes \mathbf{v}^{l_2})$. Importantly, these learnable parameters will not destory the equivariance of each path. However, they are limited in capturing directional information. In equivariant models, the original CG tensor product primarily captures directional information. We have previously mentioned our replacement of the CG tensor product with learnable modules. It is worth noting that our focus lies on the CG coefficients rather than the learnable parameters in the e3nn implementation.

### B.1.6 Gate Activation and Normalization

**Gate Activation.** In equivariant models, the Gate activation combines two sets of group representations. The first set consists of scalar steerable vector ($l = 0$), which are passed through standard

activation functions such as sigmoid, ReLU and SiLU. The second set comprises higher-order steerable vector (($l > 0$)), which are multiplied by an additional set of scalar steerable vector that are introduced solely for the purpose of the activation layer. These scalar steerable vector are also passed through activation functions.

**Normalization.** Normalization is a technique commonly used in neural networks to normalize the activations within each layer. It helps stabilize and accelerate the training process by reducing the internal covariate shift, which refers to the change in the distribution of layer inputs during training.

The normalization process involves computing the mean and variance across the channels. In equivariant normalization, the variance is computed using the root mean square value of the L2-norm of each type-$l$ vector. Additionally, this normalization removes the mean term. The normalized activations are then passed through a learnable affine transformation without a learnable bias, which enables the network to adjust the mean and variance based on the specific task requirements.

### B.2 Relationship Between Expressive Power and Equivariant Operations

In (Dym & Maron, 2021), Theorem 2 establishes the universality of equivariant networks based on the TFN structure:

**Theorem.** For all $n \in \mathbb{N}$, $\mathbf{l}_T \in \mathbb{N}_+{}^*$,

- 1. For $D \in \mathbb{N}_+$, every G-equivariant polynomial $p : \mathbb{R}^{3 \times n} \to W_{\mathbf{1}_T}^n$ of degree $D$ is in $F_{C(D),D}^{TFN}$.

- 2. Every continuous G-equivariant function can be approximated uniformly on compact sets by functions in $\cup_{D \in \mathbb{N}_+} F_{C(D),D}^{TFN}$.

Here, $n$ represents the number of input points (or nodes), $\mathbf{l}_T$ represents the degree of the approximated G-equivariant function, $C$ represents the number of channels, and $D$ represents the degree of the TFN (Tensor Field Network) structure, which is equivalent to the term $l$ used in our method. The TFN structure consists of two layers, including convolution and self-interaction. Self-interaction involves equivariant linear functions. The convolution operation calculates the CG tensor product between different steerable representations, which is a fundamental operation for transforming directional information. Most equivariant models based on group representations use a similar approach (CG tensor product) to capture directional features. Therefore, the theorem mentioned above also applies to building blocks based on CG tensor products, such as SEGNN (Brandstetter et al., 2022) and Equiformer (Liao & Smidt, 2023).

It is important to note that achieving an infinite degree in practice is not feasible. However, equivariant models based on group representations can enhance their expressive power by increasing the number of maximal degrees (Dym & Maron, 2021). In their evaluation of expressive power, as presented in (Joshi et al., 2023), the authors utilize the GWL (geometric Weisfeiler-Leman) graph isomorphism test. In Table 2 of their work, it is evident that equivariant models with a maximal degree denoted as $L$ are incapable of distinguishing $n$-fold symmetric structures when $n$ exceeds the value of $L$.

## C Proofs and Details For Section 3

### C.1 Proof for Equivariance of EST

The proof for Theorem 1 is shown in the following.

*Proof.* Recall that FT can be represented by

$$f(\vec{\mathbf{p}}) = \sum_{l=0}^{L} \sum_{m=-l}^{l} x^{(l,m)} Y^{(l,m)}(\vec{\mathbf{p}}) = (\mathbf{x}^{(0 \to L)})^T \mathbf{Y}^{(0 \to L)}(\vec{\mathbf{p}}). \tag{29}$$

When we apply a random rotation $\mathbf{D}^{-1}$ to $\mathbf{x}$, we obtain

$$(\mathbf{D}^{-1}\mathbf{x}^{(0 \to L)})^T \mathbf{Y}^{(0 \to L)}(\vec{\mathbf{p}}) \tag{30}$$

$$= (\mathbf{x}^{(0 \to L)})^T \mathbf{D}\mathbf{Y}^{(0 \to L)}(\vec{\mathbf{p}}) \tag{31}$$

$$= (\mathbf{x}^{(0 \to L)})^T \mathbf{Y}^{(0 \to L)}(\mathbf{R}\vec{\mathbf{p}}) \tag{32}$$

$$= f(\mathbf{R}\vec{\mathbf{p}}), \tag{33}$$

where $\mathbf{R}$ and $\mathbf{D}$ share the same transformation parameters.

Given a spatial representation $f(\vec{\mathbf{p}})$ transformed from a steerable representation $\mathbf{x}$, spherical attention can be represented as:

$$\tilde{f}(\vec{\mathbf{p}}_1) = \int_{\vec{\mathbf{p}}_2 \in \Omega} a\big(f^Q(\vec{\mathbf{p}}_1), f^K(\vec{\mathbf{p}}_2)\big) f^V(\vec{\mathbf{p}}_2) d\vec{\mathbf{p}}_2, \tag{34}$$

where $a(\cdot)$ denotes the operation to compute attention coefficients. Terms $Q, K, V$ denote three independent linear transformations. When the origin representation $\mathbf{x}$ is rotation by a Wigner-D matrix $\mathbf{D}^{-1}$, the representation $f(\vec{\mathbf{p}})$ is transformed to $f(\mathbf{R}\vec{\mathbf{p}})$. Therefore, the attention results are changed to

$$\int_{\mathbf{R}\vec{\mathbf{p}}_2 \in \Omega} a\big(f^Q(\mathbf{R}\vec{\mathbf{p}}_1), f^K(\mathbf{R}\vec{\mathbf{p}}_2)\big) f^V(\mathbf{R}\vec{\mathbf{p}}_2) d\mathbf{R}\vec{\mathbf{p}}_2$$

$$= \int_{\mathbf{R}\vec{\mathbf{p}}_2 \in \Omega} \frac{\exp\big(f^Q(\mathbf{R}\vec{\mathbf{p}}_1) * f^K(\mathbf{R}\vec{\mathbf{p}}_2)\big)}{\int_{\mathbf{R}\vec{\mathbf{p}}_3 \in \Omega} \exp\big(f^Q(\mathbf{R}\vec{\mathbf{p}}_1) * f^K(\mathbf{R}\vec{\mathbf{p}}_3)\big) d\mathbf{R}\vec{\mathbf{p}}_3} f^V(\mathbf{R}\vec{\mathbf{p}}_2) d\mathbf{R}\vec{\mathbf{p}}_2$$

$$= \int_{\mathbf{R}\vec{\mathbf{p}}_2 \in \Omega} \frac{\exp\big(f^Q(\mathbf{R}\vec{\mathbf{p}}_1) * f^K(\mathbf{R}\vec{\mathbf{p}}_2)\big)}{\int_{\vec{\mathbf{p}}_3 \in \Omega} \exp\big(f^Q(\mathbf{R}\vec{\mathbf{p}}_1) * f^K(\vec{\mathbf{p}}_3)\big) d\vec{\mathbf{p}}_3} f^V(\mathbf{R}\vec{\mathbf{p}}_2) d\mathbf{R}\vec{\mathbf{p}}_2 \tag{35}$$

$$= \int_{\vec{\mathbf{p}}_2 \in \Omega} \frac{\exp\big(f^Q(\mathbf{R}\vec{\mathbf{p}}_1) * f^K(\vec{\mathbf{p}}_2)\big)}{\int_{\vec{\mathbf{p}}_3 \in \Omega} \exp\big(f^Q(\mathbf{R}\vec{\mathbf{p}}_1) * f^K(\vec{\mathbf{p}}_3)\big) d\vec{\mathbf{p}}_3} f^V(\vec{\mathbf{p}}_2) d\vec{\mathbf{p}}_2$$

$$= \int_{\vec{\mathbf{p}}_2 \in \Omega} a\big(f^Q(\mathbf{R}\vec{\mathbf{p}}_1), f^K(\vec{\mathbf{p}}_2)\big) f^V(\vec{\mathbf{p}}_2) d\vec{\mathbf{p}}_2$$

$$= \tilde{f}(\mathbf{R}\vec{\mathbf{p}}_1).$$

Therefore, the output of spherical attention remains steerable after rotation. If we transform $\tilde{f}(\mathbf{R}\vec{\mathbf{p}}_1)$ into its steerable representation, denoted as $\tilde{\mathbf{x}}$, the following relationship holds:

$$\tilde{\mathbf{x}} = \mathbf{D}^{-1}\mathcal{F}^{-1}\big(\mathrm{SA}(\mathcal{F}(\mathbf{x}))\big) = \mathcal{F}^{-1}\big(\mathrm{SA}(\mathcal{F}(\mathbf{D}^{-1}\mathbf{x}))\big) \tag{36}$$

The equivariance of spherical attention holds. Moreover, the spherical FFNs is obviously equivariant because the function $\tilde{f}(\vec{\mathbf{p}}_1) = \mathrm{FFN}\big(f(\vec{\mathbf{p}}_1)\big)$ only focus on one orientation. Therefore, we can prove that the whole EST framework contained spherical attention and spherical FFN is equivariant.

In the derivation above, the rotational term is effectively omitted in the integration limits because the sphere is a symmetric set that is invariant under any rotation. Specifically, the continuous spherical surface $\Omega$ satisfies $\mathbf{R}\Omega \equiv \Omega$ for any $\mathbf{R} \in SO(3)$. This implies that for any direction $\vec{\mathbf{p}} \in \Omega$, the rotated point $\mathbf{R}\vec{\mathbf{p}}$ remains in $\Omega$, and this mapping is bijective. Since the Jacobian determinant of a rotation matrix is 1, the integral measure is invariant, i.e., $d(\mathbf{R}\vec{\mathbf{p}}) = d\vec{\mathbf{p}}$. Thus, integrating over the rotated coordinates yields the same result as integrating over the original coordinates.

To illustrate this, consider a 2D analog: a periodic function on a circle $y = f(\theta)$. The integral over the circle is $\int_0^{2\pi} f(\theta)\,d\theta$. If we rotate the domain by a constant $C$, the function becomes $f(\theta + C)$. Due to the periodicity of the circle ($f(2\pi + \theta) = f(\theta)$), the integral $\int_0^{2\pi} f(\theta + C)\,d\theta$ yields the same result as the original integral. The 3D spherical case follows the same principle: rotation changes the relative positions of points, but the integral over the entire closed surface remains invariant.

**Conclusion of Proof:** Since $\tilde{f}(\mathbf{R}\vec{\mathbf{p}}_1)$ is the rotated output, if we transform it back into its steerable representation $\tilde{\mathbf{x}}$, the following relationship holds:

$$\tilde{\mathbf{x}} = \mathbf{D}^{-1}\mathcal{F}^{-1}\big(\mathrm{SA}(\mathcal{F}(\mathbf{x}))\big) = \mathcal{F}^{-1}\big(\mathrm{SA}(\mathcal{F}(\mathbf{D}^{-1}\mathbf{x}))\big). \tag{37}$$

Thus, the equivariance of spherical attention holds. Moreover, the spherical FFN is inherently equivariant because the function $\tilde{f}(\vec{\mathbf{p}}_1) = \mathrm{FFN}\big(f(\vec{\mathbf{p}}_1)\big)$ operates locally on each orientation. Therefore, the entire EST framework is equivariant. $\qquad\square$

**Discussion on Finite Sampling and Equivariance:** Theorem 1 establishes the theoretical correctness of EST, proving strict equivariance on the continuous sphere (infinite sampling points). However, practical implementation requires discrete sampling.

In the discrete case, the attention operation in equation 35 is approximated as:

$$\tilde{f}(\vec{\mathbf{p}}_1) = \sum_{\vec{\mathbf{p}}_2} a\big(f^Q(\vec{\mathbf{p}}_1), f^K(\vec{\mathbf{p}}_2)\big) f^V(\vec{\mathbf{p}}_2). \tag{38}$$

After introducing random rotations, this becomes:

$$\tilde{f}(\mathbf{R}\vec{\mathbf{p}}_1) = \sum_{\vec{\mathbf{p}}_3} a\big(f^Q(\mathbf{R}\vec{\mathbf{p}}_1), f^K(\vec{\mathbf{p}}_3)\big) f^V(\vec{\mathbf{p}}_3). \tag{39}$$

Strict equivariance relies on the closure of the sampling point set under the rotation group. While the continuous sphere is closed under any rotation $\mathbf{R} \in SO(3)$, a finite set of sampling points cannot achieve full closure for arbitrary rotations. Closure is only guaranteed for specific rotations where every rotated point $\mathbf{R}\vec{\mathbf{p}}_i$ maps exactly to another point in the original set $\vec{\mathbf{p}}_j$.

For example, if the points are perfectly uniformly distributed and symmetric, the set forms a closed group under a specific subset of rotations. In this ideal scenario, the summation in Eq. equation 39 is identical to Eq. equation 38 (up to permutation), preserving strict equivariance. However, for arbitrary rotations not covered by the symmetry of the sampling grid, strict equivariance is compromised.

To mitigate this, we employ approximately uniform sampling methods, such as Fibonacci Lattices. Furthermore, as established in prior literature, neural networks are capable of approximating continuous equivariant functions. Rotations not strictly covered by the discrete sampling grid are handled via training, allowing the model to learn approximate equivariance. Our experiments confirm that the equivariance error reduces significantly after training without requiring explicit data augmentation.

**Sampling strategies** Most previous works use the e3nn implementation for spherical Fourier transform. However, it significantly destroy the uniformity of spherical sampling, which is illustrated in Figure 2(a). In contrast, Fibonacci Lattices (FL) do not directly divide the polar angle and azimuth angle into a grid. Instead, they select sampling points on the sphere in a spiral pattern. As shown in Figure 2(b), FL tends to achieve more uniform sampling, thereby improving the equivariance of EST. This is also consistent with the results observed in Table 6.

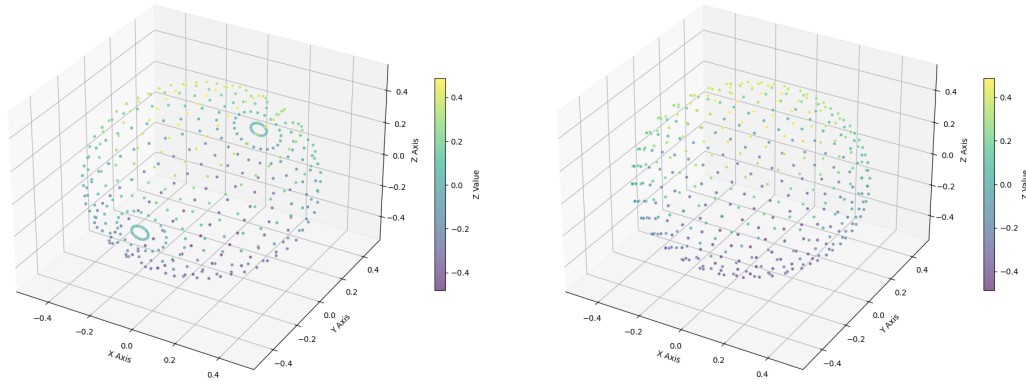

(a) e3nn Sampling          (b) Fibonacci Lattices Sampling

Figure 2: Two spherical sampling strategies.

**Enhanced Uniformity** Building on the FL sampling, we further propose a scattering optimization strategy similar to molecular dynamics. We define a distance-based interaction force between every two points: the greater the distance, the smaller the force. Then, we iterate two processes: 1. Simultaneously optimize the three-dimensional coordinates of all points through the interaction forces; 2. Re-project all points back onto the sphere. After multiple rounds of iteration, the spherical sampling points can become more uniform. It is worth noting that both the FL sampling and the dynamics simulation are conducted during the model initialization phase. Once we obtain the uniform sampling points, we save their results and directly read them in the model's feedforward process.

## C.2 EXPRESSIVENESS OF EST

In Proposition 2, wo claim that the function space spanned by the spherical Transformer encompasses that spanned by the CG tensor product and give an intuitive explanation. Here, we provide a mathematical explanation for further understanding.

Given two steerable vectors $\mathbf{u} \in \mathbb{V}_{0 \to l_1}$ and $\mathbf{v} \in \mathbb{V}_{0 \to l_2}$, and we define the CG tensor product result is $\mathbf{w} \in \mathbb{V}_{l_0}$, where $l_0 <= l_1 + l_2$. The spatial representations of $\mathbf{w}$ can be represented as:

$$\sum_{l_0, m_0} \mathbf{w}^{(l_0, m_0)} Y^{(l_0, m_0)} = \sum_{l_0, m_0} (\mathbf{u}^{(0 \to l_1)^T} \mathbf{C}_{(0 \to l_1),(0 \to l_2)}^{(l_0, m_0)} \mathbf{v}^{(0 \to l_2)}) Y^{(l_0, m_0)}, \tag{40}$$

where $\mathbf{C}_{(0 \to l_1),(0 \to l_2)}^{(l_0, m_0)}$ is a matrix including the whole CG coefficients corresponding to degree $l_0$ and order $m_0$. We temporarily ignore the input of the spherical harmonics. The spherical Transformer use the multiplication between spatial representations of $\mathbf{u}$ and $\mathbf{v}$:

$$\sum_{l_1, m_1} \mathbf{u}^{(l_1, m_1)} Y^{(l_1, m_1)} \sum_{l_2, m_2} \mathbf{v}^{(l_2, m_2)} Y^{(l_2, m_2)}$$

$$= \sum_{l_1, m_1, l_2, m_2} \mathbf{u}^{(l_1, m_1)} Y^{(l_1, m_1)} \mathbf{v}^{(l_2, m_2)} Y^{(l_2, m_2)} \tag{41}$$

$$= \mathbf{u}^{(0 \to l_1)^T} (\mathbf{Y}^{(0 \to l_1)^T} \mathbf{Y}^{(0 \to l_2)}) \mathbf{v}^{(0 \to l_2)}$$

Recall that CG coefficients are in fact the expansion coefficients of a product of two spherical harmonics in terms of a single spherical harmonic (see equation 18):

$$Y^{(l_1, m_1)} Y^{(l_2, m_2)} = \sum_{l_0, m_0} \sqrt{\frac{(2l_1 + 1)(2l_2 + 1)}{4\pi(2l_0 + 1)}} \mathcal{C}_{(l_1, m_1)(l_2, m_2)}^{(0,0)} \mathcal{C}_{(l_1, m_1)(l_2, m_2)}^{(l_0, m_0)} Y^{l_0, m_0}, \tag{42}$$

equation 41 can be transformed to

$$\mathbf{u}^{(0 \to l_1)^T} (\mathbf{Y}^{(0 \to l_1)^T} \mathbf{Y}^{(0 \to l_2)}) \mathbf{v}^{(0 \to l_2)}$$

$$= \sum_{l_0, m_0} \mathbf{u}^{(0 \to l_1)^T} \mathbf{H}_{(0 \to l_1),(0 \to l_2)}^{(l_0, m_0)} \mathbf{v}^{(0 \to l_2)} Y^{(l_0, m_0)}, \tag{43}$$

where $\mathbf{H}_{(l_1, m_1),(l_2, m_2)}^{(l_0, m_0)} = \sqrt{\frac{(2l_1+1)(2l_2+1)}{4\pi(2l_0+1)}} \mathcal{C}_{(l_1,m_1)(l_2,m_2)}^{(0,0)} \mathcal{C}_{(l_1,m_1)(l_2,m_2)}^{(l_0,m_0)}$. Compared to $\mathbf{C}_{(0 \to l_1),(0 \to l_2)}^{(l_0, m_0)}$, the term $\mathbf{H}_{(0 \to l_1),(0 \to l_2)}^{(l_0, m_0)}$ introduces an additional constant term that can be approximated by linear layers and the FFNs. Therefore, equation 41 is equivalent to equation 40. On the other hand, equation 41 is inherently part of the Transformer architecture: when relationships between different orientations are masked and the $\mathbf{V}$ vectors are taken as constant vectors, the Transformer can naturally reduce to equation 41.

Through the above process, we find that the spherical representation of the output from the CG tensor product is fundamentally a special case of EST. Furthermore, *EST can capture dependencies between different orientations, which is particularly beneficial for approximating higher degree spherical harmonic representations. For instance, at a orientation $(\theta, \varphi)$, higher-degree spherical harmonics may involve contributions from other orientations such as $(2\theta, \varphi), (\theta, 2\varphi), (2\theta, 2\varphi)$ (see $Y^{(2)}$ in equation 20). The Transformer in EST can directly combine these lower-degree terms in these orientations to approximate the higher-order terms at the orientation $(\theta, \varphi)$.*

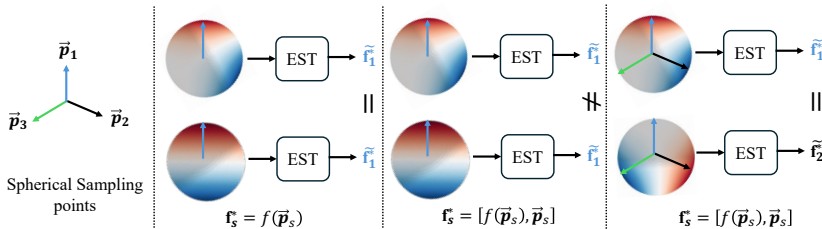

Figure 3: An example of relative orientation embedding. (left) and (middle) represent locally restructured spherical functions, which should theoretically have different representations. (right) represents globally rotated spherical functions, which should theoretically have the same (rotated) representation.

We propose an alternative interpretation of the tensor product and spherical harmonics. The traditional tensor product can combine steerable vectors of any degree combination, and its functional form is bilinear. When we transform these steerable vectors to the spatial domain, the information from the different vectors, regardless of their degree, is projected onto a sphere. This projection allows a Transformer structure on a sphere to effectively capture the dependencies between the input steerable vectors. Furthermore, the Spherical Fourier Transform ($\mathcal{F}$) and its inverse ($\mathcal{F}^{-1}$) can be expressed as two matrix multiplications: $\mathcal{F}(\mathbf{x}) = W_{FT}\mathbf{x} = \mathbf{f}^*$ and $\mathcal{F}^{-1}(\mathbf{f}^*) = W_{IFT}\mathbf{f}^* = \mathbf{x}$. Here, $W_{FT}$ and $W_{IFT}$ are matrices of size $S \times (L+1)^2$ and $(L+1)^2 \times S$, respectively, which contain the Fourier basis. The vectors $\mathbf{x}$ and $\mathbf{f}^*$ are the steerable representation and spherical representation, with sizes $(L+1)^2 \times 1$ and $S \times 1$.

## C.3 EQUIVARIANCE OF RELATIVE ORIENTATION EMBEDDING

The equivariance of relative orientation embedding can be easily proven with the same way in equation 35, where $a\big(f^Q(\mathbf{R}\vec{\mathbf{p}}_1), f^K(\mathbf{R}\vec{\mathbf{p}}_2)\big)$ is transformed to

$$\frac{\exp\big(f^Q(\mathbf{R}\vec{\mathbf{p}}_1) * f^K(\mathbf{R}\vec{\mathbf{p}}_2) + (\mathbf{R}\vec{\mathbf{p}}_1)^T(\mathbf{R}\vec{\mathbf{p}}_2)\big)}{\int_{\mathbf{R}\vec{\mathbf{p}}_3 \in \Omega} \exp\big(f^Q(\mathbf{R}\vec{\mathbf{p}}_1) * f^K(\mathbf{R}\vec{\mathbf{p}}_3) + (\mathbf{R}\vec{\mathbf{p}}_1)^T(\mathbf{R}\vec{\mathbf{p}}_3)\big)d\mathbf{R}\vec{\mathbf{p}}_3}. \tag{44}$$

Due to the spherical symmetry, we can still eliminate the rotation $\mathbf{R}$ acting on $\mathbf{p}_2$ and $\mathbf{p}_3$. Therefore, after incorporating the Relative Orientation Embedding, EST retains its equivariance.

We further elaborate on the contribution of the relative orientation embedding. In Figure 3(left), EST without positional encoding cannot distinguish locally restructured spherical functions, but it can after using positional encoding (Figure 3(middle)). In Figure 3 (right), the relative positional encoding ensures the overall equivariance of the spherical functions.

# D DETAILS OF MoE

## D.1 MIXTURE OF EXPERTS IN LANGUAGE MODELS

Recently, empirical evidence consistently demonstrates that increased model parameters and computational resources yield performance gains in language models when sufficient training data is available. However, scaling models to extreme sizes incurs prohibitive computational costs. The Mixture-of-Experts (MoE) architecture has emerged as a promising solution to this dilemma. By enabling parameter scaling while maintaining moderate computational requirements, MoE architectures have shown particular success when integrated with Transformer frameworks. These implementations have successfully scaled language models to substantial sizes while preserving performance advantages.

## D.2 MIXTURE OF EXPERTS FOR TRANSFORMERS

We begin with a standard Transformer language model architecture, which comprises $Y$ stacked Transformer blocks:

$$\mathbf{u}^y = \text{Self-Att}\left(\mathbf{h}^{m-1}\right) + \mathbf{h}^{m-1}, \tag{45}$$

$$\mathbf{h}^y = \text{FFN}\left(\mathbf{u}^y\right) + \mathbf{u}^y, \tag{46}$$

where $\text{Self-Att}(\cdot)$ denotes the self-attention module, $\text{FFN}(\cdot)$ denotes the Feed-Forward Network (FFN), $\mathbf{u}^y$ are the hidden states of all tokens after the $y$-th attention module, and $\mathbf{h}^y \in \mathbb{R}^d$ is the output hidden state after the $y$-th Transformer block. layer normalization is omitted for brevity. The MoE architecture substitutes FFN layers in Transformers with MoE layers and each MoE layer comprises multiple structurally identical experts. Each token is dynamically assigned to several experts based on learned routing probabilities: If the $y$-th FFN is substituted with an MoE layer, the computation for its output hidden state $\mathbf{h}^y$ can be expressed as:

$$\mathbf{h}^y = \sum_{i=1}^{N} \left(g_i \, \text{FFN}_i\left(\mathbf{u}^y\right)\right) + \mathbf{u}^y, \tag{47}$$

$$g_i = \begin{cases} s_i, & s_i \in \text{Topk}(\{s_j | 1 \le j \le N\}, K), \\ 0, & \text{otherwise}, \end{cases} \tag{48}$$

$$s_i = \text{Softmax}_i\left((\mathbf{u}^y)^T \mathbf{e}_i^y\right), \tag{49}$$

where $N$ denotes the total number of experts, $\text{FFN}_i(\cdot)$ is the $i$-th expert FFN, $g_i$ denotes the gate value for the $i$-th expert, $s_i$ denotes the token-to-expert affinity, $\text{Topk}(\cdot, K)$ denotes the set comprising $K$ highest affinity scores among those calculated for all $N$ experts, and $\mathbf{e}_i^y$ is the learnable parameters representing the centroid of the $i$-th expert in the $y$-th layer. The sparsity property ($K \ll N$) ensures computational efficiency by restricting each token to interact with only $K$ experts.

### D.3    MIXTURE OF HYBRID EXPERTS IN EST

Inspired by MoE of language models, we developed the mixture of hybrid experts for EST. Its computation can be expressed as:

$$\tilde{\mathbf{m}} = \sum_{i=1}^{N_{steerable}} \left(g_i \, \text{SteerableFFN}_i\left(\mathbf{m}\right)\right) + \sum_{j=1}^{N_{spherical}} \left(g_j \, \text{SphericalFFN}_i\left(\mathbf{m}\right)\right) + \mathbf{m}, \tag{50}$$

$$g_i = \begin{cases} s_i, & s_i \in \text{Topk}(\{s_k | 1 \le k \le N_{steerable}\}, K), \\ 0, & \text{otherwise}, \end{cases} \tag{51}$$

$$g_j = \begin{cases} s_j, & s_j \in \text{Topk}(\{s_k | 1 \le k \le N_{spherical}\}, K), \\ 0, & \text{otherwise}, \end{cases} \tag{52}$$

$$s_i, s_j = \text{split}\left(\text{Softmax}\left(\mathbf{m}^{(0)}\mathbf{W}\right)\right), \tag{53}$$

where $\mathbf{m}$ and $\tilde{\mathbf{m}}$ denote the input and output message, respectively, $\mathbf{m}^{(0)}$ denotes the invariant part of message, $\mathbf{W} \in \mathbb{R}^{C \times (N_{steerable} + N_{spherical})}$ represents the learnable expert centroids. Here, we omit the symbol of layer order for simplicity.

### D.4    INTEGRATION TO EQUIVARIANT ARCHITECTURES

EST with MoE can be employed both in the message block to compute edge features and in the update block to refine node embeddings. Figure 4(b,c) defines a message passing layer with EST. When computing the message, the $\mathbf{Q}$ in SA come from the aggregation of node $i$ and $j$ embeddings, while $\mathbf{K}$ and $\mathbf{V}$ is derived from the spherical harmonic representation of the relative position $\vec{\mathbf{r}}_{ij}$. In the SA of the update block, $\mathbf{Q}$, $\mathbf{K}$, and $\mathbf{V}$ are all derived from the aggregated message. Additionally, in the message and update blocks shown in Figure 4, the spatial expert is used after spherical attention, and the steerable expert is employed as a parallel branch. They are combined using gate weights.

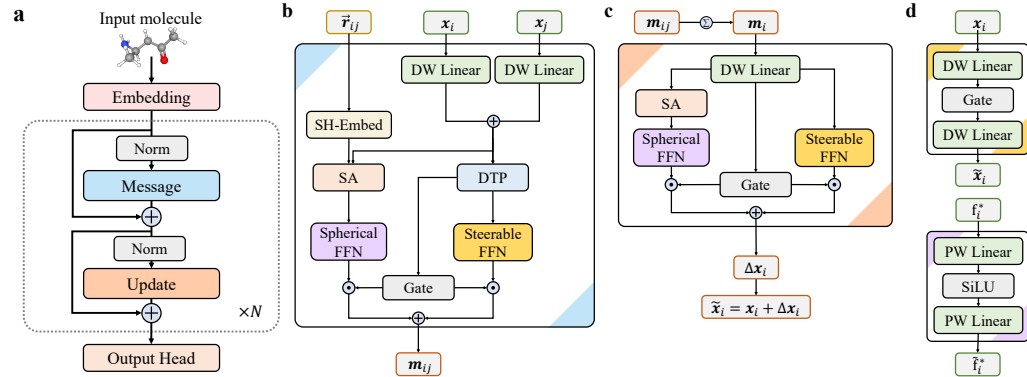

Figure 4: **The architecture and building blocks of EST.** SH and DTP denote spherical harmonic embedding and depth-wise tensor product (Liao & Smidt, 2023), respectively. For simplicity, the Fourier and inverse Fourier transform steps are omitted. (a) Overall architecture. (b) Message block. (c) Update block. (d) Two experts operating on the steerable and spatial representations, respectively.

### D.5 ABLATION STUDIES

In terms of MoE, there are three key hyper-parameters, including expert count, routing sparsity (top-K), and load balancing. Our ablation studies begin with the easiest steerable MoE using the QM9 dataset and Equiformer baseline.

- Expert count: We assessed model performance across expert counts ranging from 1 to 64. Performance improved up to 10 experts, reaching optimal results at that point (Table 7). Meanwhile, excessive expert count leads to a significant increase in time cost while the improvement in performance may not necessarily continue. In contrast, setting the number of expert as 10 offers the right balance between performance improvement and time cost.

- Routing Sparsity: We evaluated different levels of sparsity by varying the K values in the top-K router, while increased sparsity (lower K values) degraded performance (Table 8).

- Load Balancing: The auxiliary balance loss was added to the training loss with varying ratios (0 to 0.1). However, introducing balance loss consistently reduced model performance (Table 9).

The discrepancy between our findings and typical language model observations regarding sparsity and balancing can be attributed to several factors. While sparsity and balancing mechanisms generally enhance efficiency in standard language models, the limited size of the QM9 dataset plays a crucial role in this divergence. Introducing more experts increases model complexity, which can lead to overfitting rather than improved generalization, particularly when training data is scarce. Furthermore, given our limited number of experts, the necessity for stringent sparsity enforcement and elaborate load balancing mechanisms is diminished. In such scenarios, employing a dense MoE configuration tends to yield optimal performance.

Table 7: Performance and time cost depending on expert count when predicting $\alpha$ on QM9.

| Number of experts | 1(w/o MoE) | 2 | 4 | 8 | 10 | 16 | 32 | 64 |
|---|---|---|---|---|---|---|---|---|
| Test MAE $(\text{bohr}^3)$ | 0.0466 | 0.0461 | 0.0503 | 0.0437 | **0.0424** | 0.0449 | 0.0438 | 0.0443 |
| Time per epoch $(s)$ | 373.69 | 391.93 | 416.53 | 472.80 | 500.71 | 576.01 | 893.20 | 1664.93 |

Table 8: Effect of routing sparsity for predicting $\alpha$ on QM9.

| K | 2 | 3 | 5 | 10 (Dense) |
|---|---|---|---|---|
| Test MAE $(\text{bohr}^3)$ | 0.04338 | 0.04349 | 0.04472 | **0.0424** |

Table 9: Impact of balance loss ratio for predicting $\alpha$ on QM9.

| Ratio | 0 (w/o balance loss) | 0.001 | 0.01 | 0.1 |
|---|---|---|---|---|
| Test MAE $(\text{bohr}^3)$ | **0.0424** | 0.04369 | 0.04369 | 0.04361 |

# E    DETAILS OF EXPERIMENTS AND SUPPLEMENTARY EXPERIMENTS

## E.1    IMPLEMENTATION DETAILS OF BASELINES

Table 10: Hyper-parameters for the EST model setting on OC20 S2EF and OC20 IS2RE experiments.

| Hyper-parameters | S2EF-ALL | IS2RE |
|---|---|---|
| Optimizer | AdamW | AdamW |
| Learning rate scheduling | Cosine learning rate with linear warmup | Cosine learning rate with linear warmup |
| Warmup epochs | 0.01 | 2 |
| Maximum learning rate | $4 \times 10^{-4}$ | $2 \times 10^{-4}$ |
| Batch size | 256 | 32 |
| Number of epochs | 4 | 20 |
| Weight decay | $1 \times 10^{-3}$ | $1 \times 10^{-3}$ |
| Dropout rate | 0.1 | 0.2 |
| Energy coefficient $\lambda_E$ | 4, 2 | 1 |
| Force coefficient $\lambda_F$ | 100 | - |
| Gradient clipping norm threshold | 100 | 100 |
| Model EMA decay | 0.999 | 0.999 |
| Cutoff radius (Å) | 12 | 5 |
| Maximum number of neighbors | 20 | 500 |
| Number of radial bases | 600 | 128 |
| Dimension of hidden scalar features in radial functions | 128 | 64 |
| Maximum degree $L_{max}$ | 6 | 1 |
| Maximum order $M_{max}$ | 2 | 1 |
| Number of Layers | 8 | 6 |
| Node embedding dimension | 128 | $(256, l=0), (128, l=1)$ |
| Intermediate dimension during the Fourier transform | 128 | 256 |
| Intermediate dimension and the number of steerable FFN | 128, 6 | $[(768, l=0), (384, l=1)], 5$ |
| Intermediate dimension and the number of spherical FFN | 512, 4 | 512, 5 |
| Number of spherical point samples | 128 | 128 |

In the S2EF experiment, the results of baselines in Table 1 follow Liao et al. (2024b), where each model is trained in official configuration. Most of these configurations can be found in Fairchem repository, and we also follow its code framework to construct our OC20 experiments. In the IS2RE experiment, the results in Table 2 of baselines follow Liao & Smidt (2023) and Zitnick et al. (2022). In the QM9 experiment, the results in Table 3 of baselines follow Liao et al. (2024b) and Aykent & Xia (2025).

## E.2    IMPLEMENTATION DETAILS OF EST EXPERIMENTS

**S2EF and IS2RE**   In our experiments on OC20, we adopt two hybrid models: S2EF combines the message module from EquiformerV2 with the update module of EST, while IS2RE combines the message module of Equiformer with the update module of EST. To ensure a fair comparison, all training configurations are aligned with those of the original EquiformerV2 and Equiformer. The hyperparameters specific to the EST architecture include the number of spherical sampling points, the number and dimension of experts in the steerable FFN, and the number of experts in the spherical FFN. Detailed configurations for all models are summarized in Table 10. The experiment on S2EF is conducted on 32 NVIDIA A100 GPUs and the experiment on S2EF is conducted on 8 NVIDIA A100 GPUs.

**QM9**   In the QM9 experiments, we design two EST variants: 1) a fully EST-based architecture with both message and update blocks; 2) a hybrid model combining the message block of Equiformer and the update block of EST . To ensure a fair comparison with state-of-the-art methods (Equiformer and EquiformerV2 ), we adopt similar configurations as shown in Table 11. Note that Equiformer employs two different configurations for the following properties: $\alpha, \Delta\varepsilon, \varepsilon_{\text{HOMO}}, \varepsilon_{\text{LUMO}}, \mu, C_v, R^2$, ZPVE, and $G, H, U, U_0$. We follow the same strategy in our experiments. Comparisons with another SOTA model, GotenNet Aykent & Xia (2025), are conducted under different configurations, and the details are provided in Section 12.

Table 11: Hyper-parameters for QM9 dataset.

| Hyper-parameters | EST-based model (Figure 4) | EST (with GA) |
|---|---|---|
| Optimizer | AdamW | AdamW |
| Learning rate scheduling | Cosine learning rate with linear warmup | Cosine learning rate with linear warmup |
| Warmup steps | 5 | 5 |
| Maximum learning rate | $5 \times 10^{-4}, 2 \times 10^{-4}$ | $5 \times 10^{-4}, 1.5 \times 10^{-4}$ |
| Batch size | $128, 64$ | $128, 64$ |
| Max training epochs | $350, 700$ | $300, 600$ |
| Weight decay | $5 \times 10^{-3}, 0$ | $5 \times 10^{-3}, 0$ |
| Dropout rate | $0.0, 0.2$ | $0.0, 0.2$ |
| Number of radial bases | 128 for Gaussian radial basis 8 for radial bessel basis | |
| Cutoff radius (Å) | 5 | 5 |
| $L_{max}$ | 2 | 2 |
| Number of layers | 6 | 6 |
| Node dimension | 128 | $(128, l = 0), (64, l = 1), (32, l = 2)$ |
| Spherical harmonics embedding dimension | $(1, l = 0), (1, l = 1), (1, l = 2)$ | $(1, l = 0), (1, l = 1), (1, l = 2)$ |
| Intermediate dimension during the Fourier transform | 128 | 128 |
| Intermediate dimension and the number of steerable FFN | $5, 128$ | $6, (768, l = 0), (384, l = 1)$ |
| Intermediate dimension and the number of spherical FFN | $5, 512$ | $4, 512$ |
| Number of spherical point samples | $200, 128$ | 128 |

Table 12: Hyper-parameters for the EST (with GATA) models on QM9 experiments.

| Hyper-parameters | EST (with GATA) | |
|---|---|---|
| properties | $\alpha, \Delta\varepsilon, \varepsilon_{HOMO}, \varepsilon_{LUMO}, \mu, C_v, R^2$ | $G, H, U, U_0, U$, ZPVE |
| Loss function | MAE | MSE |
| Warmup steps | 0 | 10000 |
| Weight decay | 0 | 0.01 |
| Intermediate dimension and the number of steerable FFN | 512, 1 | 512, 4 |
| Intermediate dimension and the number of spherical FFN | 512, 1 | 512, 4 |
| Learning rate scheduling | linear warmup with reduce on plateau | |
| Optimizer | AdamW | |
| Maximum learning rate | $1 \times 10^{-4}$ | |
| Batch size | 32 | |
| Max training epochs | 1000 | |
| Dropout rate | 0.1 | |
| Number of RBFs | 64 | |
| Cutoff radius (Å) | 5 | |
| Lmax | 2 | |
| Number of layers | 4 | |
| Node dimension | 256 | |
| Edge dimension | 256 | |
| Number of attention heads | 8 | |
| Number of spherical point samples | 64 | |

### E.3 SUPPLEMENTARY EXPERIMENTS

**Different expert configurations** To further understand the contributions of EST and MoE, we conducted experiments on OC20 IS2RE and QM9 using various expert configurations. $(x + y)$ denotes $x$ steerable experts and $y$ spherical experts. As shown in Table 13 and Table 14, we have several findings: 1.The update block using only spherical attention still achieves improvements. The minimal MoE configuration (1 steerable expert and 1 spherical attention expert) achieves results close to the multi-expert configuration.

Table 13: The results of different expert configurations on OC20 IS2RE dataset.

| Model | energy (ID) MAE | energy (OOD Ads) MAE | energy (OOD Cat) MAE | energy (OOD Both) MAE | average |
|---|---|---|---|---|---|
| Equiformer (1 + 0) | 504 | 688 | 521 | 630 | 586 |
| Equiformer + EST (5 + 5) | 501 | 652 | 502 | 578 | 558 |
| Equiformer + EST (0 + 1) | 507 | 664 | 513 | 593 | 569 |
| Equiformer + EST (1 + 1) | 502 | 657 | 504 | 588 | 563 |

Table 14: Results of different expert configurations on QM9 dataset.

| Model | $\alpha$ | $\varepsilon_{HOMO}$ | $U_0$ |
|---|---|---|---|
| Equiformer (1 + 0) | 0.046 | 15 | 6.59 |
| Equiformer + EST (4 + 6) | 0.041 | 14 | 5.64 |
| Equiformer + EST (0 + 1) | 0.045 | 15 | 6.85 |
| Equiformer + EST (1 + 1) | 0.042 | 14 | 5.73 |
| Equiformer + EST (10 + 0) | 0.043 | 15 | 6.47 |
| Equiformer + EST (8 + 0) | 0.044 | 15 | 6.41 |
| Equiformer + EST (4 + 0) | 0.050 | 15 | 6.68 |
| Equiformer + EST (2 + 0) | 0.046 | 15 | 6.74 |

**Building Blocks** We conducted ablation studies to validate the effectiveness of individual building blocks within EST. Specially, in Table 15, removing the steerable FFN means all experts are replaced by the spherical FFN, removing the spherical FFN means all experts are replaced by the steerable FFN and removing FL Sampling refers to using the Fourier transform from e3nn instead. We used the prediction of the $\alpha$ property on QM9 as the core task for evaluation. As shown in Table 15, all components positively contribute to the overall performance. We draw several conclusions:

1. The **SA module with layer normalization** effectively improves overall performance.

2. **Mixing steerable and spherical experts** helps strike a balance between equivariance and expressive power, leading to better generalization performance.

3. **FL Sampling is crucial for EST**. Disrupting equivariance without it significantly harms the results on QM9.

Table 15: Ablation studies for modules in HDGNN.

| Building blocks in EST | | | | | $\alpha$ MAE |
|---|---|---|---|---|---|
| LayerNorm | SA | Steerable FFN | Spherical FFN | FL Sampling | $bohr^3$ |
| - | ✓ | ✓ | ✓ | ✓ | 0.046 |
| - | - | ✓ | ✓ | ✓ | 0.044 |
| ✓ | ✓ | - | ✓ | ✓ | 0.045 |
| - | - | ✓ | - | ✓ | 0.043 |
| ✓ | ✓ | ✓ | ✓ | - | 0.053 |
| ✓ | ✓ | ✓ | ✓ | ✓ | **0.041** |

**Results on Mptrj dataset** We further evaluated EST on the Mptrj dataset Deng et al. (2023), which involves predicting energies, atomic forces and stress in molecular dynamics trajectories. As shown in Table 17, EST outperforms other baseline models, demonstrating its effectiveness in modeling dynamic molecular systems. The detailed experimental settings are provided in Table 16. These results demonstrate that EST is effective in complex molecular systems. We use the combination of

EST + EquiformerV2-S with a smaller batch size compared to (Barroso-Luque et al., 2024) (256 vs. 512). For fair comparison, we reproduced EquiformerV2-S, which already surpasses the reported results in (Barroso-Luque et al., 2024). Crucially, our EST-enhanced results even outperform the force and stress metrics achieved by the larger model and the auxiliary task DeNS in (Barroso-Luque et al., 2024) (eqV2-L+DeNS). This experiment provides strong further validation of the high expressive power of our EST.

Table 16: Hyper-parameters for Mptrj dataset.

| Hyper-parameters | EquiformerV2 | EST |
|---|---|---|
| Optimizer | AdamW | AdamW |
| Learning rate scheduling | Cosine learning rate with linear warmup | Cosine learning rate with linear warmup |
| Warmup epochs | 0.1 | 0.1 |
| Warmup factor | 0.2 | 0.2 |
| Maximum learning rate | $4 \times 10^{-4}$ | $4 \times 10^{-4}$ |
| Batch size | 512, 256 | 256 |
| Number of epochs | 100 | 100 |
| Weight decay | $1 \times 10^{-3}$ | $1 \times 10^{-3}$ |
| Model EMA decay | 0.999 | 0.999 |
| Number of radial bases | 512 for Gaussian radial basis | |
| Cutoff radius (Å) | 6 | 6 |
| $L_{max}$ | 4 | 4 |
| Number of layers | 8 | 8 |
| Node dimension | 128 | 128 |
| Energy loss coefficient | 5 | 5 |
| Force loss coefficient | 20 | 20 |
| Isotropic stress loss coefficient | 5 | 5 |
| Anisotropic stress loss coefficient | 5 | 5 |
| Intermediate dimension during the Fourier transform | | 512 |
| Intermediate dimension and the number of steerable FFN | | 128, 4 |
| Intermediate dimension and the number of spherical FFN | | 128, 4 |
| Number of spherical point samples | | 72 |

Table 17: Results on Mptrj dataset. DeNS (Liao et al., 2024a) is an auxiliary training task.

| model | energy (meV/atom) ↓ | forces (meV/Å) ↓ | stress (meV/$^3$) ↓ | forces cos ↑ |
|---|---|---|---|---|
| Units | $bohr^3$ | meV | meV | meV |
| EquiformerV2-S | 12.4 | 32.22 | 1.55 | 0.72 |
| EquiformerV2-S + DeNS | 11.43 | 31.67 | 1.44 | 0.72 |
| EquiformerV2-M + DeNS | 11.17 | 31.46 | 1.48 | 0.728 |
| EquiformerV2-L + DeNS | 10.58 | 30.48 | 1.47 | 0.738 |
| EquiformerV2-S (Repetition) | 11.1 | 29.48 | 1.39 | 0.74 |
| EquiformerV2-S + EST | 10.78 | 28.87 | 1.32 | 0.74 |

