# OpenReview forum: "Equivariant Spherical Transformer for Efficient Molecular Modeling"
_ICLR.cc/2026/Conference — Submitted to ICLR 2026_

### Official Review · Reviewer_mddm · 2025-10-24

**Soundness:** 2
**Presentation:** 3
**Contribution:** 2
**Rating:** 2
**Confidence:** 5

**Summary:**

The paper introduces the Equivariant Spherical Transformer (EST), a novel framework for 3D molecular modeling. The authors posit that existing Equivariant Graph Neural Networks (GNNs), which heavily rely on Clebsch-Gordan (CG) tensor product convolutions, suffer from limited non-linearity and expressiveness. EST aims to overcome this by operating in the Fourier spatial domain. The method works by transforming steerable group representations into a sequence of points on a sphere, then applying a Transformer-based architecture to model dependencies between these points.

The core architectural ideas are promising. I am open to increasing my score during the rebuttal phase if the authors can provide the necessary clarifications and more rigorous comparisons to fully substantiate their claims against other SOTA models.

**Strengths:**

1. The theoretical analysis (Theorem 1) and empirical validation (Table 5) of how spherical sampling uniformity (Fibonacci lattice vs. e3nn grid) impacts $SO(3)$ equivariance is a valuable and rigorous contribution
2. The "hybrid mixture of experts"  is a thoughtful design choice, allowing the model to explicitly balance the strict equivariance of steerable FFNs with the enhanced expressive power of spherical FFNs.
3. The model demonstrates very strong results on the large-scale and challenging OC20 benchmark, outperforming several larger models and those trained with additional data.

**Weaknesses:**

1.  The paper's most significant weakness is its flawed handling of GotenNet, a key state-of-the-art baseline that also avoids CG transforms.
    - The justification provided, that GotenNet uses an "extensive training schedule", is methodologically unsound. It's well-established that different network architectures may require different training schedules (epochs, batch sizes, optimizers) to converge optimally. This, by itself, is not a valid academic reason to exclude a major baseline from a SOTA comparison.
    - Furthermore, if the authors' unstated concern was computational cost, "extensive schedule" (i.e., 1000 epochs with a 32 batch size ) is not a valid proxy for this. The relevant metric is total wall-clock training/inference time. A model with a lower per-epoch cost could be more efficient overall, even with more epochs. The authors fail to provide this crucial efficiency comparison, making it impossible to evaluate their claims of efficiency and performance against a top competitor.

2. The comparison that _is_ provided in Appendix E.3 (Table 11) is flawed and does not constitute a fair evaluation.
    - The authors compare against the 4-layer GotenNet, which is the smallest, weakest, and known-to-be-underparameterized version of that model. The original GotenNet paper's main results clearly show significant performance gains from its 6-layer (Base) and 12-layer (Large) models, which are conveniently ignored here.
    - The comparison model, misleadingly named "EST (with GATA)", is not a clean comparison. It is, in fact, the full 4-layer GotenNet architecture (using its initialization, GATA message block, and EQFF) where the authors have added their spherical FFN component to run in parallel with GotenNet's existing EQFF block . This is a (GotenNet + EST components) hybrid, not a direct EST vs. GotenNet comparison.  The negligible performance difference shown in Table 11 on an already weak baseline demonstrates little, if any, benefit from the EST addition and can likely be attributed to a simple increase in parameters.
    - The authors could have simply trained their proposed EST model with the "extensive schedule" for a direct comparison. The choice to instead create a convoluted, modified 4-layer GotenNet hybrid strongly suggests that the standard EST model, when trained under an equivalent schedule, did not achieve comparable SOTA results.

3. Given that both architectures are transformer-based and use attention (GATA in GotenNet), a much more insightful and direct comparison would have been to integrate the proposed spherical attention/relative orientation embedding _directly into_ GotenNet's GATA module. This would have tested the EST's core components as a _replacement_ or _enhancement_ to an existing equivariant attention mechanism, rather than an appendage to a different block (EQFF).
4. The "plug-and-play" claim is undermined by the OC20 experiments. The paper's SOTA results on OC20 (Table 1, Table 2) are not from the full EST architecture (Figure 3). Instead, they use a hybrid model where the EST's message block is replaced by the message block from Equiformer/EquiformerV2. This makes it impossible to isolate the contribution of the novel EST update block. The strong performance could be largely attributed to the borrowed Equiformer message passing, not the novel spherical transformer.

**Questions:**

1. What is the performance (MAE) of the *full* EST architecture (using its native message block from Figure 3b) on the OC20 benchmarks, as opposed to the hybrid model reported in Tables 1 & 2?

2. To quantify the "hybrid mixture of experts" trade-off, what is the MAE and latency (ms/batch) when using *only* steerable FFN experts versus *only* spherical FFN experts?

3. How do the authors reconcile the model's strong theoretical expressiveness on synthetic n-fold symmetry tasks (Table 4) with its practical performance on the QM9 benchmark (Table 3), which is not SOTA on all tasks?

4. What is the practical trade-off between the number of sampling points ($S$), model accuracy (MAE on QM9), and latency (ms/batch)?

5. Could the authors provide end-to-end training and inference latency (ms/batch) for EST against *all* key SOTA baselines from Table 3, including Equiformer, EquiformerV2, and GotenNet, using identical hardware and batch sizes?

---

> ### Author Response · Authors · 2025-11-23
>
> Thank you for your valuable review. We have revised the manuscript as your suggestion, highlighting the changes in **blue** in the revision.
>
> **In response to your greatest concern regarding the flawed handling of GotenNet, we have supplemented a series of experiments during the rebuttal phase and added detailed explanations in the QM9 section of the main text.** To put it simply, we did indeed overlook the excellent performance of GotenNet in terms of efficiency and performance. We have added experiments for the base and large versions and trained the full EST framework for the same number of time steps. On the QM9 benchmark, GotenNet still achieves a leading level of performance. We have also provided an analysis of efficiency and performance in the revised paper.
>
> After carefully examining the GotenNet model, we find some model difference need to be noted: **EST and GotenNet contribute research of equivariant models in different directions**: 1. GotenNet focuses on explicitly learning the interactions between neighboring atoms, thereby efficiently capturing local geometric features or rich interatomic interactions. 2. EST enhances the upper limit of the expressive power of the basic unit, enabling the model to learn some deep physical mechanisms from node embeddings.
>
> Moreover, **we have also addressed your concerns regarding the “plug-and-play” module and explained why we focus on using EST in the Update block (Responses to weaknesses 3&4).**
>
> # Responses to weaknesses 1
> Thank you for your reminder. **We have now comprehensively compared the computational costs of EST and GotenNet in the main text and provided some analysis of the results.** To summarize, we acknowledge that "different network architectures require different training schedules" and have included GotenNet in the main table for comparison. We have also provided latency of training and inference. GotenNet indeed demonstrates higher training and inference efficiency. Moreover, the additional computational overhead introduced by EST when used as a plug-in is within an acceptable range. For Equiformer, six steerable experts and four spherical experts increase latency by about 21%. For GotenNet, one steerable expert and one spherical expert increase latency less that 20%.
>
> We still want to emphasize **the performance improvements that EST has achieved in some non-saturated tasks, such as the results of Equiformer on QM9 and those in OC20.** You can refer to our "Responses to weaknesses 1" to reviewer Ls5n. The performance improvement of EST is not just due to the increase in parameters. As we said, generalization performance is a balance between equivariance and expressiveness. EST is more suitable for tasks that require strong expressiveness, such as complex molecular systems. Specifically, EST has achieved better performance on OC20 S2EF than methods using more training data and more model parameters.
>
> # Responses to weaknesses 2
> Thank you for your reminder. **We have comprehensively compared all models of GotenNet and also used pure EST.** These contents can be found in the experiment section of the main text. To sum up, EST does not bring much improvement to the base and large models. We have analyzed this phenomenon and consider that the slight loss of equivariance in EST limits the generalization performance. **The final generalization performance of the model is a balance between equivariance and expressiveness. QM9 is a relatively small dataset with higher requirements for equivariance, and GotenNet has already achieved a performance close to saturation. In this case, the loss of equivariance will have a more obvious impact on generalization performance.**
>
> # Responses to weaknesses 3
> **The attention mechanism of GotenNet is based on neighbors, learning geometric features by calculating the weights of edge features. In contrast, EST computes the weights for each component of the spatial embedding without crossing different edges or nodes.** This makes EST more like a feature filter for individual embeddings. Our approach is different from various previous equivariant attention mechanisms. Therefore, we prefer to directly apply it to node embeddings to extract deep features, instead of capturing dependencies between edges or nodes.

---

> ### Author Response · Authors · 2025-11-23
>
> # Responses to weaknesses 4
> Sorry for the misunderstanding. **Our “plug-and-play” refers to using EST as a universal module that can be quickly integrated into existing equivariant frameworks to enhance performance.** This term is quite common in some visual models, such as [1,2]. The biggest feature of a plug-and-play module is that it can be directly applied to existing advanced architectures, avoiding cumbersome hyperparameter design. Secondly, as we said in “Responses to weaknesses 3”, EST is a basic computational unit for individual embeddings and does not explicitly consider the interactions between edges or nodes, which makes it independent of specific architectures.
>
> [1] ConvFormer: Plug-and-Play CNN-Style Transformers for Improving Medical Image Segmentation
>
> [2] Squeeze-and-Excitation Networks
>
> There are three reasons for applying EST to the Update block: 1. The high expressiveness of EST is more suitable for extracting key information from complex features. Therefore, we choose to use it to process the input of the Update block, which is the aggregated message. 2. The Update block consumes fewer computational resources, enhancing the scalability of the MoE mechanism. 3. The Update block of most equivariant models consists of node-wise “linear layers + nonlinear activation layers + linear layers” structure. This structure can naturally serve as steerable experts for MoE, making it easy to integrate EST into the Update block.
>
> # Responses to questions 1
> We apologize that we are unable to supplement the OC20 S2EF experiments during the rebuttal period, as it is a massive dataset and our computing resources are limited. We will endeavor to add the OC20 IS2RE experiments. If we can complete the training during the rebuttal period, we will notify you on OpenReview. If not, we will store the results in an anonymous repository. We have explained in "Responses to weaknesses 3" and "Responses to weaknesses 4" why we focus on using EST in the update. In addition, we have added some experiments based on the number of experts (see our "Responses to weaknesses 1" to reviewer Ls5n), which can assist you in evaluating the performance of EST.
>
> # Responses to questions 2
> On QM9, we compared models using only 1 steerable expert and only 1 spherical expert. The latency is shown in the table below ((ms/batch)). The batch sizes of all models are set to 32 (one Nvidia H800 Gpu).
>
> || Equiformer steerable | Equiformer spherical | GotenNet-S  steerable | GotenNet-S spherical  | GotenNet-L  steerable | GotenNet-L spherical  |
> | -| -| -|-|-|-|-|
> | Train latency (ms/batch) | 185 | 182 | 55 |60 | 116 | 122 |
> | Inference latency (ms/batch) | 67 | 69 | 35 | 39 | 106 | 113 |
>
> # Responses to questions 3
> The generalization performance of molecular models is usually related to two key properties: equivariance and expressiveness. Among them, the expressiveness of GNN describes the theoretical upper limit of the model, that is, what level of equivariant functions can be represented. The equivariant functions that different physical properties depend on are different. For example, the Coulomb interaction can be represented as a characteristic with $ l = 1 $. Equivariant models usually determine the maximum order $ L $ based on the complexity of the molecular system. For example, in QM9, the $ L $ of most models is less than 4, but for OC20, $ L \geq 6 $ will achieve better results. This also reflects that the organic small molecules in QM9 have less dependence on higher-order physical interactions. The higher-order expressiveness brought by EST may be redundant. On the other hand, learning equivariance is difficult with small datasets. The lack of equivariance can cause the model to overfit to specific rotations, which is also a reason why EST does not achieve stable improvements on QM9.
>
> # Responses to questions 4
> Good question! If the number of sampling points is too small, it will lead to information loss in the Fourier transform. Therefore, we strictly follow the Nyquist sampling rate, i.e., $ S \geq (2L)^2 $. In actual training, we find that sampling up to $ S = (2L)^2 $ can achieve good performance. A larger $ S $ has a very weak impact on performance but will bring greater latency. We provided a complexity analysis in our "response to Question 1" from reviewer Ls5n, which you can refer to as supplementary information.
>
> # Responses to questions 5
> We have provided the training and inference latency in the experiments section of revised version (Table 4).

---

> > ### Author Response · Authors · 2025-12-03
> > **Thanks**
> >
> > We greatly appreciate your detailed and constructive review. In our first-round rebuttal, we promised to supplement the experiments of the full EST model (EST message block + EST update block) on the OC20 IS2RE benchmark. Fortunately, we have successfully completed the experiments and observed that the EST model still achieves SOTA results, as shown in the table below. We can see that the results for Pure EST are very similar to those of the Equiformer message block + EST update block combination. This demonstrates that applying the plug-and-play EST building block to the update block is an effective strategy.
> >
> > | Model                                   | energy (ID) MAE | energy (OOD Ads) MAE | energy (OOD Cat) MAE | energy (OOD Both) MAE | average |
> > | --------------------------------------- | --------------- | -------------------- | -------------------- | --------------------- | ------- |
> > | Equiformer (Pure Equiformer)    | 504             | 688                  | 521                  | 630                   | 586     |
> > | Equiformer + EST (Paper result) | 501             | 652                  | 502                  | 578                   | 558     |
> > | Pure EST | 508             | 648                  | 505                  | 574                   | 555     |

---

### Official Review · Reviewer_qRcS · 2025-10-29

**Soundness:** 3
**Presentation:** 3
**Contribution:** 3
**Rating:** 6
**Confidence:** 4

**Summary:**

The authors propose Equivariant Spherical Transform (EST), a new "plug-and-play" framework that applied a Transformer-like architecture to the Fourier spatial domain of group representations in order to achieve higher expressivity than tensor-product based equivariant model. EST preserves equivariance and achieves good performance on OC20 and QM9 benchmarks. The authors further provide theoretical support for their central claims about EST.

**Strengths:**

### Strengths
- The authors provide a good explanation of the Spherical Fourier Transform and of their proposed method. The writing is overall easy to follow and understand.
- The fact that EST is more expressive than tensor-product-based models is a very strong and novel contribution
- EST can be integrated into existing model designs, making is a flexible approach that can build on existing works

**Weaknesses:**

### Weaknesses
- The authors do not have a very convincing set of experiments. OC20 and QM9 are older and relatively saturated datasets, and most recent works on MLIPs are training and evaluating on the SPICE-MACE-OFF [1] and MPtrj datasets [2]. The paper is also missing comparisons to many recently developed MLIPs such as eSEN [3].
- The authors do not attempt to train a larger scale "foundation" model based on EST or provide ablation experiments to demonstrate the scaling of the proposed method. While EST is a strong theoretical contribution, it may be the case that expressivity doesn't matter when the model is sufficiently large/training on diverse data. The paper would be more convincing if EST could outperform SOTA foundation models.


[1] MACE-OFF: Transferable Short Range Machine Learning Force Fields for Organic Molecules, Kovács et al. (2023), https://arxiv.org/abs/2312.15211

[2] CHGNet: Pretrained universal neural network potential for charge-informed atomistic modeling, Deng et al. (2023), https://arxiv.org/abs/2302.14231

[3] Learning Smooth and Expressive Interatomic Potentials for Physical Property Prediction, Fu et al. (2025), https://arxiv.org/abs/2502.12147

**Questions:**

### Questions (related to above weaknesses)
- Can the authors train EST on MPtrj and evaluate on Matbench Discovery?
- Can the authors train a large model(s) to demonstrate the scalability of EST?

---

> ### Author Response · Authors · 2025-11-23
>
> Thank you for your valuable review. We have revised the manuscript as your suggestion, highlighting the changes in **blue** in the revision.
>
> # Responses to weaknesses 1
>
> Thanks for your suggestions. **We're working hard to train the EST model on MPtrj. Apologies for the slow progress due to limited resources. If we finish the training during the rebuttal period, we'll inform you directly on OpenReview and update the paper. If not, we'll upload the results to an anonymous repository and revise the paper in the future.**
>
> Thanks for pointing out the eSEN comparison. However, it doesn't have official results on the benchmarks in our paper. We'll include it as a comparison baseline after finishing the MPtrj training.
>
> Lastly, we acknowledge that the QM9 and OC20 datasets are older and relatively saturated. But we'd like to emphasize that the OC20 S2EF in our paper involves a complex molecular system with ample training samples. Its training results are stable and it's still widely used for evaluating advanced equivariant models. **The comparisons we made on OC20 are not favorable to EST.we only trained the model on the S2EF All training set.  However, its performance has already surpassed the base model trained with more data (All + MD). In addition, we also compare with the models only trained with S2EF All, EST achieves better results compared to models with more layers (8 layers vs. 20 layers) and parameters (45M vs. 153M).**
>
> To ensure the comprehensiveness of the paper, we have already cited the work you mentioned in the introduction of the revised version.
>
> # Responses to weaknesses 2
> We apologize that, due to resource limitations, we are unable to extensively train large-scale "foundation" models. We acknowledge your point that "it may be the case that expressivity doesn't matter when the model is sufficiently large/training on diverse data." However, this is predicated on the model having basic equivariance and expressive power. For instance, we could use a large language model to learn molecular systems[1]. It can approximate any continuous equivariant function and exhibits strong expressive power. But due to the complete lack of equivariance, such models typically rely on extremely large amounts of training data or data augmentation to function effectively. The greatest contribution of EST is that it expands expressive power while maintaining the vast majority of equivariance. It is also meaningful for research that is not related to foundation models.
>
> [1] Transformers Adaptively Learn Molecular Structures Without Graph Priors
>
> # Responses to question 1
>
> We are working hard to train an EST on the MPtrj dataset. Fingers crossed that we can get it done by the rebuttal period. In addition, you may also want to take another look at our OC20 experiments, where our EST achieved better results in an unfair comparison. More details can be found in Responses to weaknesses 1.
>
> # Responses to question 2
>
> We are sorry that we didn't have enough resources to train large foundational models during the rebuttal period. We will make up for this in the future. We also emphasize that the main contribution of EST is to enhance general equivariant neural networks and inspire the design of equivariant architectures, not limited to training large foundational models.

---

> > ### Comment · Reviewer_qRcS · 2025-11-27
> > **Response to Author Comments**
> >
> > I thank the authors for the response. I will maintain my score and recommend acceptance of this paper. I also recommend the authors to include the additional results on the MPtrj dataset in the revised manuscript.

---

> > > ### Author Response · Authors · 2025-12-03
> > > **Thanks and Responses to Mptrj Benchmark**
> > >
> > > We sincerely thank you for the positive assessment of our revised manuscript and for recommending acceptance. We promised to supplement the Mptrj experiments in our first round of responses. Fortunately, **we have successfully completed the EST experiments, and they show a significant improvement, as presented in the table below.**
> > >
> > > | Model                     | energy (meV/atom) $\downarrow$ | forces (meV/$\AA$) $\downarrow$ | stress (meV/$\AA^3$) $\downarrow$ | forces cos $\uparrow$ |
> > > | ------------------------- | ------------------------------ | ------------------------------- | --------------------------------- | --------------------- |
> > > | eqV2-S (results from [1])    | 12.4                           | 32.22                           | 1.55                              | 0.72                  |
> > > | eqV2-S+DeNS (results from [1])    | 11.43                          | 31.67                           | 1.44                              | 0.72                  |
> > > | eqV2-M+DeNS (results from [1])      | 11.17                          | 31.46                           | 1.48                              | 0.728                 |
> > > | eqV2-L+DeNS (results from [1])   | 10.58                          | 30.48                           | 1.47                              | 0.738                 |
> > > | EqV2-S (reproduced by us) | 11.1                           | 29.42                           | 1.39                              | 0.74                  |
> > > | EqV2-S+EST (Ours)         | 10.78                          | 28.87                           | 1.32                              | 0.74                  |
> > >
> > > DeNS [2] is an auxiliary training task, not the building block like EST. S, M and L denote Small, Middle and Large.
> > >
> > > These results demonstrate that EST is effective in complex molecular systems. We use the combination of EST + EquiformerV2-S with a smaller batch size compared to [1] (256 vs. 512). For fair comparison, we reproduced EquiformerV2-S, which already surpasses the reported results in [1]. Crucially, our EST-enhanced results even outperform the force and stress metrics achieved by the larger model and the auxiliary task DeNS in [1] (eqV2-L+DeNS). This experiment provides strong further validation of the high expressive power of our EST.
> > >
> > > We provide the detailed experiments on MPtrj in Appendix E.3. We thank you again for your suggestions.
> > >
> > > [1] Open Materials 2024 (OMat24) Inorganic Materials Datasetand Models
> > >
> > > [2] Generalizing Denoising to Non-Equilibrium Structures Improves Equivariant Force Fields

---

### Official Review · Reviewer_Ls5n · 2025-10-29

**Soundness:** 2
**Presentation:** 2
**Contribution:** 2
**Rating:** 2
**Confidence:** 4

**Summary:**

The authors introduce a new architecture for equivariant modeling. They begin by presenting the concepts of spherical and spatial representations of functions defined on a sphere. Prior works such as SCN and ESCN apply only pointwise nonlinearities on grids, which limits computations to individual nodes. This paper proposes a novel self-attention mechanism that defines attention across pairs of nodes, thereby enhancing expressiveness. To preserve equivariance, the authors introduce a uniform spherical sampling strategy for the attention computation. Building on these ideas, they develop a new architecture that achieves good empirical results on OC20 and QM9.

**Strengths:**

1. The proposed theoretical framework presents valuable insights that could substantially contribute to the future development of equivariant architectures.

**Weaknesses:**

1. The primary weakness lies in the insufficient experimental evaluation. Specifically, the experiments lack efficiency comparisons against existing architectures such as Equiformer, eSEN, and Equiformer V2. Additionally, the introduction of the mixture-of-hybrid-experts module appears orthogonal to the proposed theoretical framework, which diminishes the perceived contribution of the spherical attention mechanism. It gives the impression that the theoretical innovation provides limited practical gains, while most of the performance improvement stems from unrelated components. To strengthen the paper, the authors should introduce minimal modifications to an established architecture and compare the efficiency/performance of the proposed method within that controlled setting. For instance, a variant such as “Equiformer + Spherical Attention” would offer a clearer assessment of the method’s effectiveness.
2. Lack of results on the OC20 test set weakens the empirical validation. Moreover, Table 1 is difficult to interpret and confusing; the authors should follow the formatting used in the EscAIP paper by clearly separating training (All or All+MD) and test sets (val or test) to improve readability and comparability.
3. The proof of the main theorems in Appendix C.1 lacks clarity. The authors should justify the drop of the rotation terms in Equation (34), lines 3 and 4, and explain how this leads to problems when the formulation is discretized.
4. Theorem 1 asserts strict SO(3) equivariance under the condition that the sample set P is closed under arbitrary rotations. However, for any finite P, this condition cannot hold except in trivial cases: only the continuous sphere satisfies closure, whereas a finite grid does not. The paper instead relies on the notion that “uniform sampling approximates closure,” which yields **approximate**, rather than strict, equivariance. The authors should revise this claim accordingly.

Minor Problems:
1. Double citation on Gotennet.
2.    In Eq. (11), you define $z_s = \frac{2s-1}{S-1}$ and then use $\sqrt{1-z_s^2})$. For $(s=S), (z_s>2)$, which is outside [-1,1]. I believe you intended something like $z_s = 1 - \frac{2s-1}{S}) (or (\frac{2s}{S}-1)$. Please fix this; otherwise sampling is ill-defined.

**Questions:**

1. What is the time-complexity of the proposed spherical attention in terms of $L$?
2. Can you provide a bound relating sampling discrepancy to equivariance error?

---

> ### Author Response · Authors · 2025-11-23
>
> Thank you for your valuable review. We have revised the manuscript as your suggestion, highlighting the changes in **blue** in the revision.
>
> # Responses to weaknesses 1
>
> **(Computational Efficiency)** We first address your concern about efficiency, we report the model throughput in Table 1. It can be observed that **the throughput of EST (8 layers) is very close to that of the base model EquiformerV2 (8 layers) (6.8 vs. 7.1), and significantly exceeds the throughput of the deeper EquiformerV2 (20 layers) (6.8 vs. 1.8). It is also worth noting that EST (8 layers) outperforms EquiformerV2 (20 layers) in terms of energy and forces prediction.**
>
> We also evaluated the training efficiency of EST during the rebuttal period. We used the OC20 S2EF experiments (using Nvidia H800 GPUs). As shown in the table below, the impact of EST on training is greater than on inference, but still within an acceptable range.
>
> || Training Time per step | Memory per GPU |
> | -| -| -|
> | EquiformerV2 | 0.5180 s | 66570 MiB|
> | EST+EquiformerV2  | 0.7425 s | 66600 MiB|
>
> The latency comparison on QM9 can be found in Table 4 of the revision.
>
> **(Ablation study for MoE)** We understand your concerns regarding the mixture-of-hybrid-experts module and have supplemented the corresponding experiments. In these experiments, **we used only spherical attention experts (0+1) or one steerable equivariant expert plus one spherical attention expert (1+1) in the update block.** These ablation experiments were based on the OC20 IS2RE experiments. As shown in the table below, we have several findings: 1.The update block using only spherical attention still achieves improvements, with the average MAE decreasing from 586 to 569. The minimal MoE configuration (1 steerable expert and 1 spherical attention expert) achieves results very close to the multi-expert configuration (5+5) (average MAE 563 vs. 558).
>
> | Model  | ID MAE | OOD Ads MAE | OOD Cat MAE | OOD Both MAE | average |
> | -- | -- | - | - |-- | - |
> | Equiformer (1 + 0) (Pure Equiformer) | 504 | 688  | 521  | 630  | 586 |
> | Equiformer + EST (5 + 5) (Paper result) | 501 | 652 | 502 | 578 | 558 |
> | Equiformer + EST (0 + 1)  | 507 | 664 | 513 | 593 | 569 |
> | Equiformer + EST (1 + 1)  | 502 | 657 | 504 | 588 | 563 |
>
> In addition, we have supplemented the above models with ablation experiments on the QM9 dataset, as well as a version that uses only steerable experts under MoE (an extension of Equiformer, without any spherical attention). As shown in the table below, we have several findings: 1.When the impact of multiple experts is minimized (0+1 or 1+1), EST still achieves improvements. Note that QM9 is a relatively small dataset, and the lack of equivariance in EST can amplify the impact on performance. 2.If we use only the MoE strategy, there is only a stable improvement when the number of experts exceeds a certain number, and the improvement is limited compared to spherical attention experts.
>
> | Model  | $\alpha$ | $\varepsilon_{HOMO}$ | $U_0$ |
> | - | -| - | - |
> | Equiformer (1 + 0) (Pure Equiformer) | .046 | 15 | 6.59  |
> | Equiformer + EST (4 + 6) (Paper result) | .041 | 14 | 5.64 |
> | Equiformer + EST (0 + 1)  | .045 | 15 | 6.85 |
> | Equiformer + EST (1 + 1)  | .042 | 14 | 5.73 |
> | Equiformer + EST (10 + 0)  | .043 | 15 | 6.47 |
> | Equiformer + EST (8 + 0)  | .044 | 15 | 6.41 |
> | Equiformer + EST (4 + 0)  | .050 | 15 | 6.68 |
> | Equiformer + EST (2 + 0)  | .046 | 15 | 6.74 |
>
> **(Why we use MoE)** To further address your concerns, we explain the design of the mixture-of-experts (MoE) module. First, let's review the message passing framework. The message block is responsible for processing edge features, accounting for the majority of the computational load in message passing neural network. The update block handles node features, which involves less computation but needs to deal with the aggregated message from the message block, which is complex. In our work, spherical attention is a fundamental, plug-and-play equivariant learning unit, which requires it to be quickly applicable in any equivariant model without significantly reducing model efficiency. Moreover, the **high expressive power of EST makes it more capable of handling complex equivariant features.** Therefore, we choose to apply spherical attention in the update block. **To ensure its plug-and-play characteristic, we choose to add EST as a parallel branch within the update block. This is the prototype of the 1+1 MoE.** Furthermore, we find that expanding the number of parallel branches in the update block does not significantly affect inference performance, so we introduce the MoE idea. Additionally, **due to the lack of equivariance in EST, spherical attention may affect performance on some small datasets. Steerable experts can maintain equivariance but lack expressive power. Through MoE, we can find the trade-off between equivariance and expressive power in various tasks, thereby maximizing generalization performance.**

---

> ### Author Response · Authors · 2025-11-23
>
> # Responses to weaknesses 2
>
> Thank you for your suggestions. We have revised Table 1. To be brief, **due to resource limitations, we only trained the model on the S2EF All training set. However, its performance has already surpassed the base model trained with more data (All + MD).** Previous papers have also shown that almost all models achieve better performance when trained with MD data, which is in line with common sense. In addition, we also compare with the models only trained with S2EF All, EST achieves better results compared to models with more layers and parameters. Your suggestion regarding the test set is excellent, but unfortunately, we may not be able to evaluate it during the rebuttal period (the S2EF test set is very large, and our current resources are limited). However, previous methods have shown that the validation set and test set of S2EF exhibit similar results and trends (The test results of almost all models are slightly better than the validation results), and the validation set is large enough to assess the model's performance. We will supplement the results of the test set in future versions.
>
> # Responses to weaknesses 3
>
> **(Why the rotation term is dropped in our proof) The rotational term is omitted because the sphere is a symmetric set that is closed under any rotation.** In other words, the continuous spherical surface can be denoted as $\Omega$. **After applying an arbitrary rotation $\mathbf{R}$ (where $\mathbf{R} \in SO(3)$), the spherical space remains unchanged, which can be expressed as $\mathbf{R} \Omega \equiv \Omega$.** This implies that for any direction $\vec{\mathbf{p}} \_{i} \in \Omega$, we have $\mathbf{R}\vec{\mathbf{p}} \_{i} \in \Omega$, and this mapping is injective. Consequently, when integrating over the sphere, we obtain
> $$
> \int_{\vec{\mathbf{p}}\in \Omega}f(\vec{\mathbf{p}}) d\vec{\mathbf{p}} = \int_{\mathbf{R}\vec{\mathbf{p}}\in \Omega}f(\mathbf{R}\vec{\mathbf{p}}) d\mathbf{R}\vec{\mathbf{p}}.
> $$
>
> To facilitate understanding, consider the 2D analog: a circular function defined as $y = f(\theta)$. The integral over the entire circle is $\int \^{2\pi} \_{0}f(\theta) d\theta$. If we rotate the circle, the function becomes $f(\theta') = f(\theta + C)$, where $C$ is a constant. The integral then becomes $\int \^{2 \pi + C} \_{C}f(\theta') d\theta'$. Due to the periodicity of the circle ($f(2\pi + \theta) = f(\theta)$), this integral can be rewritten as
>
> $$
> \int \^{2\pi} \_{C}f(\theta') d\theta' + \int \^{C} \_{0}f(\theta') d\theta' = \int \^{2\pi} \_{0}f(\theta') d\theta',
> $$
>
> which yields the same result as the original integral. The three-dimensional case follows the same principle. Integration over the sphere essentially sums the function values over all points on the sphere; under rotation, only the ordering of these points changes, while the total sum remains invariant.
>
> We appreciate your suggestion, and we have enhanced the description of spherical integration in Appendix C.1.
>
> As mentioned above, **equivariance relies on the closure of the set under arbitrary rotations $\mathbf{R}$. However, with only a finite set of sampled points, closure can be guaranteed only for specific rotations $\mathbf{R}$ that satisfy $\mathbf{R}\vec{\mathbf{p}}_{i} \in \Omega$. Moreover, this property also depends on uniform sampling.** As illustrated in our Figure 2, the FL sampling overlaps with the original sampling under various rotations, whereas the e3nn sampling overlaps only when rotated to $180^{\circ}$ symmetric positions.

---

> ### Author Response · Authors · 2025-11-23
>
> # Responses to weaknesses 4
>
> Good question! In fact, this is also a key issue addressed in our paper. **The purpose of Theorem 1 is to explore the theoretical soundness of EST and to show that the equivariance of EST is related to the closure property of the sampling points. After presenting Theorem 1 (line 263), we pointed out that a finite set of sampling points can achieve partial closure through uniform sampling—i.e., EST is equivariant under certain specific rotations.**
>
> We also realized that we did not explicitly clarify the practical non-implementability of Theorem 1 in code, and your suggestion has helped make our revised manuscript clearer.
>
> For further discussion on finite sampling points, please refer to our response to Reviewer DiC2 under "Responses to Weaknesses 2."
>
> # Responses to Minor Problems
>
> Thank you for your careful observation, we have corrected it. The correct term in Equation 11 is $z_s=\frac{2s-1}{S}-1$.
>
> # Responses to Questions 1
>
> The complexity of the CG tensor product is related to the maximum degree $L$: $\mathcal{O}(L^6)$. In contrast, the complexity of the Transformer in EST is related to the spherical sampling number $ S $: $ \mathcal{O}(S^2) $ (self-attention). Based on the Nyquist sampling rate, the minimum number of samples on the spatial domain is at least $ (2L)^2 $, meaning the complexity of the Transformer in EST is $ \mathcal{O}(16L^4) $. The complexity of FFT and IFFT is $ 2 \cdot S \cdot L^2 \geq 8L^4 $. Therefore, **the total complexity of EST is $ \mathcal{O}(24L^4) $**. When $ L > 4 $, EST theoretically has lower computational cost. Moreover, in practical implementation, we also need to consider the impact of channel expansion on computational complexity. **Transformers can directly fuse multi-channel signals, allowing the channel dimension $ C $ in EST to be freely scaled, with complexity growing linearly with respect to $ C $: $ \mathcal{O}(24CL^4) $**. In contrast, the CG tensor product operation can only handle two single-channel irreducible representations at a time. When extending to multiple channels, if we consider pairwise interactions between all channels, the complexity increases to $ \mathcal{O}(C^2L^6) $. Some methods only consider interactions between a subset of channels (e.g., Equiformer), reducing the complexity to $ \mathcal{O}(CL^6) $, but at the cost of reduced representational capacity. In summary, EST offers greater flexibility in channel expansion while maintaining lower computational complexity.
>
> # Responses to Questions 2
>
> **(Sampling discrepancy and equivariance error).** Good question! Let's first clarify the theoretical boundary between sampling discrepancy and equivariance error: **In an ideal scenario, EST under different sampling methods can approximate strict equivariance.** This is because the Transformer on the sphere can approximate any continuous function, including continuous equivariant functions. When trained with sufficient data, it can learn the equivariance for any rotation (please refer to our response to Reviewer DiC2 under "Responses to Questions 6."). In other words, for example, **if we first train EST using a Fourier transform that approximates infinite sampling points. Then we freeze the parameters and replace the sampling points with e3nn sampling or FL sampling with a number of sampling points $S \ge (2L)^2$, the model remains equivariant for various rotations.**
>
> However, **the theoretical boundary between sampling discrepancy and equivariance error is not practically useful because we cannot achieve an approximation of infinite sampling points in training, nor can we perform data augmentation for all rotations on the data.** We need to empirically analyze the equivariance error of different sampling methods. From Table 6, we can see that FL sampling and dynamically optimized initial equivariance error are the smallest because under uniform sampling, EST is inherently equivariant for some specific rotations.

---

> ### Comment · Reviewer_Ls5n · 2025-11-27
>
> I am quite satisfied with the authors' response. I will raise my score to borderline accept.

---

> > ### Author Response · Authors · 2025-12-03
> > **Thanks**
> >
> > We are delighted that our rebuttal was largely satisfactory. We also sincerely thank you for the positive assessment of our revised manuscript and for recommending acceptance. Finally, at the conclusion of the rebuttal phase, we thank you once again for your detailed and constructive review.

---

### Official Review · Reviewer_DiC2 · 2025-10-31

**Soundness:** 2
**Presentation:** 1
**Contribution:** 2
**Rating:** 2
**Confidence:** 5

**Summary:**

The paper proposes EST (Equivariant Spherical Transformer), which applies Transformer-like attention to the Fourier spatial domain of group representations for molecular modeling. The authors claim this achieves higher expressiveness than tensor product-based methods while maintaining SO(3) equivariance through uniform spherical sampling. They evaluate on OC20 and QM9 benchmarks.

**Strengths:**

1.  Applying attention mechanisms in the spherical spatial domain is an interesting alternative to tensor product operations
2. Testing on both OC20 and QM9 with multiple metrics

**Weaknesses:**

1. Undefined Notations and Poor Presentation:
- "EST (with GA)" in Table 3 is never defined in the main text
- Multiple undefined abbreviations: EwT (Table 2), PW-Linear, DTP
- Inconsistent notation (S² vs S^2, multiple uses of C for different dimensions)
2. Theorem 1: Claims "strict SO(3)-equivariance" but this is impossible with finite sampling.
3. Fibonacci lattice sampling (Eq.~11): incorrect formula \& missing definitions. The manuscript writes the FL coordinates as
$
\vec p_s = \big[ p_1 - z_s^2 \cos(2s\pi/\lambda),\; p_1 - z_s^2 \sin(2s\pi/\lambda),\; z_s \big], \quad
z_s = \frac{2s-1}{S-1}, \quad \lambda=\frac{1+\sqrt5}{2},
$
which does not match the standard FL parametrization; the symbol $\(p_1\)$ is undefined and the $\(x,y\)$ components lack the $\(\sqrt{1-z_s^2}\)$ factor.
4. Discrete inverse SFT and quadrature weights (Eq.12) lacks mathematical justification. Eq. 12 defines
$
Y^{(l,m)*}(\vec p) = \lambda(l,m)\, Y^{(l,m)}(\vec p), \quad
\lambda(l,m) = \frac{1}{\sum_s Y^{(l,m)}(\vec p_s)^2},
$
which is unclear and nonstandard.
5. Proposition 2: The proof in Appendix C.2, where the claim that the spherical Transformer’s function space “encompasses” that of CG tensor products, is informal and insufficient.
6. The exclusion of GotenNet as a baseline from main results is highly problematic.GotenNet is reported only in Appendix E.3 via a hybrid “EST with GATA” variant, not as a main baseline. This weakens a fair comparison to a relevant SOTA method.
7. B.1.3 (relationship between spherical harmonics and Wigner-D) — incorrect equality: The appendix states
$D^{(l,m)}(R_{\alpha,\beta,\gamma}) = \sqrt{2l+1}\, Y^{(l,m)}(\vec p)$
which conflates Wigner-D matrix entries with spherical harmonics.
8. Some notation and clarity issues:
- Symbol overload: The symbol $\(R\)$ is used both for $\(\mathbb{R}\)$ reals and rotations. Please use $\(\mathbb{R}\)$ for the reals and $\(R\in\mathrm{SO}(3)\)$ (or boldface) for rotations consistently.
- Relative orientation embedding (Eq.~13): You augment queries/keys with $\(\vec p_{s}\)$. Do you normalize $\(\vec p_s\)$  before concatenation to avoid numeric scaling issues? Please state any normalization and explain how the term $\(\vec p_{s_i}^\top \vec p_{s_j}\)$ behaves under discrete sampling.
- Dimensions and shapes: Ensure the dimension counts for $\(Y^{(0\to L)}(\vec p)\)$ and the shapes in Eq.~8 and Eq.~28--32 are consistent across the manuscript.

**Questions:**

Below are explicit questions that, if answered or addressed in the revision, will substantially improve clarity, correctness, and the fairness of comparisons.
1. Fibonacci lattice (Eq.~11): Please confirm the exact FL parametrization used.
2. Nyquist/bandlimit requirement: You mention a minimum of $\((2L)^2\)$ sampling points (p.~6). Do you mean $\((L+1)^2\)$ spherical harmonic coefficients or a different bound tied to your quadrature scheme? Please clarify the precise sampling requirement as a function of $\(L\)$.
3. Why is GotenNet not included in the main comparison tables? Can you report EST with your standard message block, EST with GATA and GotenNet as separate rows in main table, and provide a short discussion on whether and how the message block choices influences performance, holding the update block constant?
4. Can you provide rigorous justification for Equation 12?
5. What is the actual computational complexity comparison with tensor products?
6. How does discretization-induced equivariance breaking affect downstream task performance?
7. Discrete equivariance error: Table~5 shows empirical equivariance errors for different sampling strategies. Can you provide a short explanation linking these errors to the sampling uniformity and the chosen $\(w_s\)$? For example, do errors decay with $\(S\)$ as expected, and how does the dynamics optimization affect the quadrature accuracy?
8 Controlled ablation with GATA: Can you add a controlled ablation that fixes the message block (GATA) and compares the update block used in GotenNet vs EST MoE (and vice versa), so readers can isolate where gains arise?

---

> ### Author Response · Authors · 2025-11-23
>
> Thank you for your valuable review. We have revised the manuscript as your suggestion, highlighting the changes in **blue** in the revision, which we agree improves clarity for the readers. Our detailed responses to all your concerns are provided below. Please do not hesitate to let us know if you have any further questions.
>
> # Responses to weaknesses 1
>
> We appreciate your suggestion. We have verified and corrected all abbreviations across the text.
> Regarding the notation for $S$ and $C$: In the initial version, we used $ s_1, s_2, \dots, s_S $ and $c_1, c_2, \dots, c_C$ to index the sampling points and channels, respectively. In the revision, we have adopted the notation $s = 1, s = 2, \dots, s = S$ and $c = 1, c = 2, \dots, c = C$. Additionally, the unit sphere is consistently denoted as $\mathbb{S}^{2}$.
>
> # Responses to weaknesses 2
>
> Your statement is correct: **Strict equivariance cannot be achieved on finite samples, which is also the focus we paid attention to after proposing Theorem 1.** To put it simply, the purpose of Theorem 1 is to explore the theoretical correctness of EST, that is, the equivariance under the continuous sphere (infinite sampling points). In addition, Theorem 1 and its proof show that **this equivariance is related to the (Group Action) closure of the sampling point set**, where the infinite point set is closed to any rotation $R\in SO(3)$. **After proposing Theorem 1 (line 265), we pointed out that finite sampling points can achieve partial closure through uniform sampling, that is, EST is equivariant for some specific rotations.**
>
> **The model will approximate the equivariance beyond specific rotations after training**: Learnable neural networks are used to approximate continuous functions, including continuous equivariant functions. Some models [1,2] also show that some learnable modules can learn approximate equivariance by adding equivariance-related constraints (rotating to local coordinate systems or using spatial domain representations). Our experiments also show that the equivariance error of the trained model will be significantly reduced (without data augmentation).
>
> Based on this weakness, **we have added further explanations of infinite sampling and finite sampling in Theorem 1 and added an discussion of equivariance loss in the appendix C.1.**
>
> [1] Spherical Channels for Modeling Atomic Interactions. C. Lawrence Zitnick et al.
>
> [2] Reducing SO(3) Convolutions to SO(2) for Efficient Equivariant GNNs. Saro Passaro et al.
>
> # Responses to weaknesses 3
>
> I think there might be some misunderstanding here. The formula in our original manuscript is $\vec{\mathbf{p}}_s = [\sqrt{1-z_s^2}\cos(2s\pi/\lambda), \sqrt{1-z_s^2}\sin(2s\pi/\lambda), z_s]$, not the you mentioned in the review (${p}_s = [p_1 -z^2_s\cos(2s\pi/\lambda), p_1 -z^2_s\sin(2s\pi/\lambda), z_s]$). Therefore, our formula does not contain the undefined symbol $p_1$, nor is it missing the $\sqrt{1-z_s^2}$ term. If I have misunderstood you, please let me know and I will make further revisions.
>
> # Responses to weaknesses 4
>
> Good question! Note that we use the Real-value Spherical harmonics (RSH). **Equation 12 is based on a mathematical theorem that the mathematical conjugate of Real-value Spherical harmonics is itself [3]**, that is, $Y \^{(l,m)*}(\vec{\mathbf{p}} \_{s})=Y \^{(l,m)}(\vec{\mathbf{p}} \_{s})$.
>
> Recall that $\int Y \^{(l,m)} Y \^{(l',m')*} d \Omega = \delta \_{ll'} \delta \_{mm'}$, where $\delta$ is Kronecker delta. In the implementation, this integral on finite sampling points may break the above formula, i.e. $\sum \_{s=1} \^{S}Y \^{(l,m)}(\vec{\mathbf{p}} \_{s})Y \^{(l,m)}(\vec{\mathbf{p}} \_{s}) \neq 1$. Therefore, Equation 12 is not our mathematical innovation, but a code trick to ensure the conjugate of RSH in finite uniform sampling.
>
> [3] https://docs.abinit.org/theory/spherical_harmonics/
>
> In the revised version, **we have added the mathematical background of Equation 12  and emphasized its contribution to the code implementation.**

---

> ### Author Response · Authors · 2025-11-23
>
> # Responses to weaknesses 5
>
> Thank you for your careful observation. We have revised the relevant statements in Proposition 2 and Appendix C.2. We used Proposition 2 instead of Theorem 2 in main text because the expressiveness of EST relies on the Transformer, which has shown a high upper boundary in many continuous function tasks but is not easy to prove rigorously.
>
> We provide an alternative way to interpret the tensor product and spherical harmonics: the traditional tensor product can combine steerable vectors of any degree combination, and its function form is bilinear. When we transform steerable vectors to the spatial domian, the information of different vectors with any degree will be projected to a sphere. Therefore, Transformer structure on sphere can capture the dependencies between input steerable vectors.
>
> In addition, Spherical Fouier Transform and its inverse form two matrix multiplications:
>
> $ \mathcal{F} (\mathbf{x}) = W \_{FT} \mathbf{x} = \mathbf{f} \^{*} $
>
> $ \mathcal{F}^{-1}(\mathbf{f} \^* ) = W \_{IFT} \mathbf{f} \^* = \mathbf{x} $
>
> , where $W \_{FT}$ and $W \_{IFT}$ with the size of $S \times (L+1)^2$ and $(L+1)^2\times S$ contain all the Fouier basis. $\mathbf{x}$ and $\mathbf{f} \^{*}$ with the size of $(L+1) \^2 \times 1$ and $S \times 1$ are steerable representation and spherical representation.
>
>
>
> In EST, the information of the steerable vector is transformed onto the spherical sampling points, and using the Transformer to capture the dependencies between different sampling points can also combine representations of different degrees. Meanwhile, the spherical Fourier transform and its inverse are implemented with twoLinear matrix multiplications. In the absence of information loss, the tensor product projected onto the spherical space remains bilinear, which is easy for the Transformer to learn.
>
> # Responses to weaknesses 6
>
> Thank you for your reminder. In the revised version, we have included GotenNet in the main text and added more detailed comparisons.
>
> # Responses to weaknesses 7
>
> The Section B.1.3 have been corrected.
>
> # Responses to weaknesses 8
>
> (1) Apologies, I couldn't find the source of this issue. We used $\mathbb{R}$ to denote the real numbers and also used $\mathbf{R}$ to represent rotation. If there are any omissions or misunderstandings, please let us know and we will be extremely grateful.
>
> (2) $\vec{\mathbf{p}} \_{s}$ is sampled from the unit sphere (Equation 11), and the norm of all $\vec{\mathbf{p}}$ is naturally normalized to 1. $\vec{\mathbf{p}} \_{s_i}^{T}\vec{\mathbf{p}} \_{s_j}$ is used to distinguish different direction pairs in the discrete state. **We have added a figure in Appendix C.3 to reflect the contribution of our relative orientation embedding.** For example, let's consider the three features $\mathbf{f} \^* \_{1},\mathbf{f} \^* \_{2}, \mathbf{f}^* \_{3}$ corresponding to three sample points $\vec{\mathbf{p}} \_{1},\vec{\mathbf{p}} \_{2},\vec{\mathbf{p}} \_{3}$. When updating the feature, self-attention will consider the contributions of all other points to this point, that is, $\hat{\mathbf{f}} \^* \_{1} = Attn(q \_{1},k \_{2}) v \_{2} + Attn(q \_{1},k \_{3}) v \_{3} + ...$. A significant problem is that if we swap the features of $\vec{\mathbf{p}} \_{2}$ and $\vec{\mathbf{p}} \_{3}$, $\hat{\mathbf{f}} \^* \_{1}$ remains unchanged, which is unreasonable because the spherical function will also change after the swap. Relative position embedding is also frequently used in NLP. For example, there are two sentences "A is behind B" and "B is behind A". They have different logic, but after processing by Transformer, the features of "A" are exactly the same, and so are the features of "B", which is fatal to understanding tasks.
>
> (3) Thank you for your suggestion. We have checked all the shapes.
>
> # Responses to questions 1
>
> Please refer to "Responses to Weaknesses 3"..
>
> # Responses to questions 2
>
> Good suggestion! The Nyquist rate refers to the minimum number of sampling points required for a Fourier transform on a sphere. The dimension in the harmonic domain is $(L+1)^2$, as it includes all steerable vectors with $l=0,1,\ldots,L$. In the implementation of the Fourier transform, we sample a point $\vec{\mathbf{p}} \_{s}$ on the sphere, and its feature $\mathbf{f} \^* \_{s}$ is obtained by taking the inner product with the $(L+1)^2$-dimensional vector $\mathbf{Y} \^{(0\to L)}(\vec{\mathbf{p}} \_{s})$. We need a total of $S \ge (2L)^{2}$ such sampling points to avoid information loss.
>
> We have added the sampling requirement function  $S \ge (2L)^{2}$ to the main text according to your suggestion.
>
> # Responses to questions 3
>
> Thank you for the suggestion. We have already included GotenNet in the main comparison table and analyzed the choice of message blocks.

---

> ### Author Response · Authors · 2025-11-23
>
> # Responses to questions 4
>
> Please refer to “Responses to weaknesses 4”. In short, Equation 12 is not mathematical innovation. It relies on the existing mathematical theorem that the conjugate of a real-valued spherical harmonic is itself. The contribution of Equation 12 is to ensure conjugacy in code implementation  through the normalization constant under finite sampling points.
>
> # Responses to questions 5
>
> The complexity of the CG tensor product is related to the maximum degree $L$: $\mathcal{O}(L^6)$. In contrast, the complexity of the Transformer in EST is related to the spherical sampling number $ S $: $ \mathcal{O}(S^2) $ (self-attention). Based on the Nyquist sampling rate, the minimum number of samples on the spatial domain is at least $ (2L)^2 $, meaning the complexity of the Transformer in EST is $ \mathcal{O}(16L^4) $. The complexity of FFT and IFFT is $ 2 \cdot S \cdot L^2 \geq 8L^4 $. Therefore, the total complexity of EST is $ \mathcal{O}(24L^4) $. When $ L > 4 $, EST theoretically has lower computational cost. Moreover, in practical implementation, we also need to consider the impact of channel expansion on computational complexity. Transformers can directly fuse multi-channel signals, allowing the channel dimension $ C $ in EST to be freely scaled, with complexity growing linearly with respect to $ C $: $ \mathcal{O}(24CL^4) $. In contrast, the CG tensor product operation can only handle two single-channel irreducible representations at a time. When extending to multiple channels, if we consider pairwise interactions between all channels, the complexity increases to $ \mathcal{O}(C^2L^6) $. Some methods only consider interactions between a subset of channels (e.g., Equiformer), reducing the complexity to $ \mathcal{O}(CL^6) $, but at the cost of reduced representational capacity. In summary, EST offers greater flexibility in channel expansion while maintaining lower computational complexity.
>
> # Responses to questions 6
>
> As we observe, the lack of equivariance has not had a significant impact on the performance of downstream tasks. First, **we show the rotation error in Table 6 and point out on line 533 that the equivariant error of the EST model trained on QM9 can reach 1.8e-5, which has a minimal impact on downstream tasks.** Second, we must note that non-equivariant models usually mitigate the risk of overfitting through data augmentation [4] or equivariance constraints [1]. The essence of many methods is to expose the model to more types of rotations so that it can learn equivariance. In Responses to weaknesses 2, we point out that the uniformly sampled EST is inherently equivariant to some rotations, which allows the model to effectively learn comprehensive equivariance without using data augmentation. We have published the training results using non-uniform sampling in Table 14 (second to last row), and we can see that the model's test set performance has declined significantly and has fallen behind the SOTA models.
>
> [4] Learning SO(3) Equivariant Representations with Spherical CNNs
>
> # Responses to questions 7
>
> First, we clarify the uniformity of the three methods: FL w/optim. > FL w/o optim. > e3nn. Among them, e3nn (see Figure 2) is a method with severe lack of uniformity. Then, we draw several conclusions about the equivariant error: 1. The equivariant error will significantly decrease with the improvement of uniformity. 2. For the FL w/o optim. method, the increase in the number of samples will also reduce the equivariant error. However, when S increases to a certain extent or the model is trained, the magnitude of the equivariant error is basically fixed. 3. In code implementation, the quadrature $\int_{\vec{\mathbf{p}} \in \Omega} Y^{(l,m)}(\vec{\mathbf{p}})Y^{(l',m')*}(\vec{\mathbf{p}})d\vec{\mathbf{p}}$ may not be strictly equal to 0 due to non-strict sampling, where $(l,m)\neq (l',m')$. Dynamics optimization can alleviate this problem.
>
> # Responses to questions 8
>
> In Section 4, we have revised the experiments on QM9 and added a comprehensive comparison with GotenNet. All GotenNet experiments fix the message block (GATA) and compare the update block used in GotenNet versus EST MoE. We have also compared the performance of models based on the EST message block with other models.

---

> > ### Comment · Reviewer_DiC2 · 2025-11-27
> >
> > Thanks for the revisions, the majority of my concerns are resolved, thus I'm raising my score

---

> > > ### Author Response · Authors · 2025-12-03
> > > **Thanks**
> > >
> > > We sincerely thank you for the positive assessment of our revised manuscript and for recommending acceptance. We greatly appreciate your detailed and constructive review.

---

### Author Response · Authors · 2025-12-03
**General response**

Dear Reviewers and Chairs,

We sincerely thank all reviewers for their constructive and insightful comments, which are invaluable for improving our manuscript. We also appreciate the Chairs for their time and efforts in coordinating the review process.

We are delighted to see the reviewers’ positive feedback on our work, noting that we have proposed a novel equivariant building block (EST) based on spherical attention and **highlighting the promising potential of our approach for the future of equivariant architectures (Reviewer Ls5n and Reviewer mddm)**. The reviewers also pointed out that **our theoretical and experimental analyses on equivariance make contributions to the interpretation of EST (Reviewer qRcS and Reviewer mddm)**. Furthermore, we appreciate the **Reviewers mddm’ recognition strong performance of EST on complex molecular systems (OC20), where our EST exceeds models with larger parameters or additional training data.** We are grateful for these recognitions.

During the rebuttal period, we responded to all the weaknesses and questions raised by the reviewers. We have summarized some common concerns:

- **The relationship between discrete sampling on the sphere and equivariance.** In our responses, we pointed out that uniform discrete sampling enables EST to be equivariant for some specific rotations, while other types of rotations can be learned during training. Table 6 in our paper also demonstrates this argument: the untrained EST already has a low equivariance error, which can be reduced to a negligible level after training. In addition, we have added explanations on the relationship between discrete sampling and equivariance in the manuscript to help readers understand more clearly.

- **Complexity.** We provided an analysis of EST's complexity in our responses. In short, the complexity of EST is $\mathcal{O}(24CL^4)$, while that of the fully connected tensor product is $\mathcal{O}(C^2L^6)$. EST shows significant advantages when $L$ is large and when channel expansion is needed.

- **Explanation for using the EST module in the update block.** Note that the proposed EST is a plug-and-play module, designed to be quickly integrated into existing equivariant models to enhance performance. Additionally, the high expressiveness of EST is more suitable for extracting key information from complex features. Therefore, we choose to use it to process the input of the Update block, which is the aggregated message. Moreover, The Update block consumes fewer computational resources, enhancing the scalability of the MoE mechanism.

- **Experiments on GotenNet.** In the first version of the manuscript, we excluded GotenNet from the core baseline and only made a simple comparison. This was our mistake. In the revised version, we have supplemented experiments for all GotenNet models + EST and included them in the core comparison for QM9. In addition, we have also added results on computational efficiency and analysis of EST's performance.

It is worth noting that, based on the suggestion of Reviewer qRcS, **we supplemented our experiments on the Mptrj benchmark, which contains a large number of complex molecular  systems. EST achieved a significant performance improvement on this benchmark, further validating its high expressive power.** The detail can be found in our final response to Reviewer qRcS and revised manuscript.

We once again sincerely thank all reviewers for their valuable feedback.

Best regards,

The Authors

---

> ### Author Response · Authors · 2025-12-03
> **Summary of Rebuttal**
>
> Dear Chairs,
>
> Thank you for handling our submission. As you step into this role, we would like to provide an additional summary along with the "General response" of the post-rebuttal status to assist your decision-making.
>
> During the discussion period, we have engaged actively with the reviewers and provided point-by-point responses to all concerns raised. We are pleased to report that this discussion has been very productive:
>
> **Reviewer DiC2** raised their rating to **6 borderline accept**, acknowledging that the majority of concerns were addressed. Their initial issues, centered on unclear exposition and mathematical representation, were resolved with clear clarifications and manuscript revisions to enhance precision and interpretability.
>
> **Reviewer Ls5n** expressed quite satisfaction with our responses and upgraded their rating to **6 borderline accept**. We successfully addressed their main concerns by adding the requested efficiency comparisons and ablation studies demonstrating the effectiveness of spherical attention components.
>
> **Reviewer qRcS** recommended acceptance and **maintained their original rating**. We have also addressed their final point by conducting new experiments on the MPtrj benchmark, confirming that EST achieves SOTA results on this challenging dataset.
>
> **Reviewer mddm.** While Reviewer mddm did not provide further discussion, their primary remaining concern was **the need for a more rigorous comparison with GotenNet.** We recall their original statement: "The core architectural ideas are promising. I am open to increasing my score during the rebuttal phase if the authors can provide the necessary clarifications and more rigorous comparisons to fully substantiate their claims against other SOTA models.”
>
> In response, we **conducted a comprehensive and rigorous comparison between EST and GotenNet model family.** In addition, we have added several ablation experiments on QM9 and OC20 to reflect the modules in which EST takes effect. In addition, we have added comparisons with the Mptrj benchmark. In short, EST achieves stable improvement in complex molecular systems and surpasses SOTA models. In small molecular systems of QM9, the advantage of EST is weakened. We have **included these experimental results in the revised version and provided the necessary clarifications, such as experimental details and analysis, the impact of imperfect variations on the saturated accuracy of QM9, and the impact of EST's strong expressive ability on different molecular systems, etc.** In addition, we also provided the requested clarification on the attention mechanism and the "plug-and-play" claim. We are confident that all of their original concerns have been fully resolved.
>
> Given the extensive improvements made during the rebuttal and the resulting support from Reviewers, we hope you will consider these positive updates favorably in your final assessment.
>
> Best regards,
>
> The Authors

---

### Meta-Review · Area_Chair_NiS2 · 2026-01-06

**Summary:**

The paper proposes Equivariant Spherical Transformer (EST), an alternative to a family of equivariant models that use Clebsch-Gordan tensor product. Instead, EST performs attention in a Fourier spatial domain.

Reviewers identified the initial version of the manuscript to have lots of issues pertaining to clarity, theoretical claims, and non-compelling experiments some of which either omitted relevant baselines or were conducted on toy saturated datasets (such as QM9).

During the rebuttal, the manuscript has been substantially changed to fix the raised issues and add more experiments - substantial enough changes to warrant another pass of reviews at a different venue.
Ultimately, the model lacks compelling experiments and has to be compared on novel and harder datasets (other than QM9 and OC20) to demonstrate practical gains of the proposed theoretical improvements, thus I recommend a reject at this stage.

**Reviewer Concerns:**

* Reviewers DiC2 and mddm raised issues about technical clarity and theoretical claims which have been addressed to a certain extent by the authors
* Reviewers Ls5n, qRcS, and mddm expressed concerns about experiments - in many of them, EST provides rather marginal improvements compared to the best existing equivariant models.

**Reviewer Scores:**

* DiC2: 2 -> 6
* Ls5n: 2 -> 6
* qRcS: no change
* mddm: no explicit decision, likely still around the borderline

---

### Decision · Program_Chairs · 2026-01-26

Reject